# Themis2 regulates natural killer cell memory function and formation

Tsukasa Nabekura [1,2,3] ✉, Elfira Amalia Deborah[2,4], Saeko Tahara[2,5,6],
Yuya Arai [2,6,7], Paul E. Love [8], Koichiro Kako[1,9], Akiyoshi Fukamizu [1],
Masafumi Muratani [10] & Akira Shibuya [1,2,3] ✉

Immunological memory is a hallmark of the adaptive immune system. Although natural killer (NK) cells are innate immune cells important for the immediate host defence, they can differentiate into memory NK cells. The molecular mechanisms controlling this differentiation are yet to be fully elucidated. Here we identify the scaffold protein Themis2 as a critical regulator of memory NK cell differentiation and function. Themis2-deficient NK cells expressing Ly49H, an activating NK receptor for the mouse cytomegalovirus (MCMV) antigen m157, show enhanced differentiation into memory NK cells and augment host protection against MCMV infection. Themis2 inhibits the effector function of NK cells after stimulation of Ly49H and multiple activating NK receptors, though not specific to memory NK cells. Mechanistically, Themis2 suppresses Ly49H signalling by attenuating ZAP70/Syk phosphorylation, and it also translocates to the nucleus, where it promotes Zfp740-mediated repression to regulate the persistence of memory NK cells. Zfp740 deficiency increases the number of memory NK cells and enhances the effector function of memory NK cells, which further supports the relevance of the Themis2-Zfp740 pathway. In conclusion, our study shows that Themis2 quantitatively and qualitatively regulates NK cell memory formation.

Immunological memory is a hallmark of the immune system[1,2]. The acquisition of immunological memory provides long-lasting host protection against infectious diseases and malignancies by eliciting improved adaptive immune responses[1,3]. However, accumulating evidence indicates that innate immune cells such as natural killer (NK) cells can also evoke memory or memory-like responses following infections or other stimulations, and these responses have been attributed to the differentiation of long-lived memory (or trained) cells with enhanced cellular functions[2,4–6]. Immunological memory is governed by the quality and quantity of memory cells, with its qualitative traits being determined by their effector function on a per-cell basis, and its quantitative traits being determined by their persistence through the balance of apoptosis and survival[7–9]. Although these traits of memory cells can be accounted for by both the activation signaling

[1]Life Science Center for Survival Dynamics, Tsukuba Advanced Research Alliance (TARA), University of Tsukuba, Ibaraki 305-8575, Japan. [2]Department of Immunology, Faculty of Medicine, University of Tsukuba, Ibaraki 305-8575, Japan. [3]R&D Center for Innovative Drug Discovery, University of Tsukuba, Ibaraki 305-8575, Japan. [4]Doctoral Program in Medical Sciences, Graduate School of Comprehensive Human Sciences, University of Tsukuba, Ibaraki 305-8575, Japan. [5]College of Medicine, School of Medicine and Health Sciences, University of Tsukuba, Ibaraki 305-8575, Japan. [6]Bioinformatics Laboratory, Faculty of Medicine, University of Tsukuba, Ibaraki 305-8575, Japan. [7]College of Biological Sciences, School of Life and Environmental Sciences, University of Tsukuba, Ibaraki 305-8575, Japan. [8]Section on Hematopoiesis and Lymphocyte Biology, Eunice Kennedy Shriver National Institute of Child Health and Human Development, National Institutes of Health, Bethesda, MD 20892, USA. [9]Faculty of Life and Environmental Sciences, University of Tsukuba, Ibaraki 305-8575, Japan. [10]Department of Genome Biology, Faculty of Medicine, University of Tsukuba, Ibaraki 305-8575, Japan. ✉e-mail: nabekura.tsukasa.fe@u.tsukuba.ac.jp; ashibuya@md.tsukuba.ac.jp

as cytoplasmic events and the transcriptional and epigenetic changes as nuclear events[10–12], the molecular machinery that regulates the differentiation into memory cells is not yet fully understood.

NK cells play an important role in anti-viral and anti-tumor immunity[13]. They recognize unhealthy cells by utilizing a repertoire of activating NK receptors and exert their cytotoxic activity and production of chemokines and cytokines such as interferon (IFN)-γ for the eradication of these cells[13,14]. NK cells have traditionally been classified as innate immune cells; however, emerging evidence has shown that NK cells can differentiate into memory cells with enhanced effector functions[5,6,9]. An activating NK receptor, Ly49H, expressed on mouse NK cells specifically recognizes the mouse cytomegalovirus (MCMV) protein m157 on infected cells and transmits an activation signal via an adaptor molecule, DAP12, and the proximal phosphorylation of ZAP70 and Syk[15–17]. In addition to the essential role of Ly49H+ NK cells in the control of MCMV infection[18], activated Ly49H+ NK cells expand as effector NK cells, undergo apoptosis after the peak of NK cell response, or differentiate into long-lived memory NK cells during MCMV infection[5,19]. Memory NK cells show several enhanced functional properties: (1) Long-term persistence, (2) the capability of undergo secondary expansion, (3) augmented cytotoxicity and IFN-γ production after stimulation of activating NK receptors in vitro, (4) provision of improved host defence against MCMV infection, and (5) enhanced anti-tumor activity in vivo[5,6,19]. Similarly, human NK cells expressing the activating NK receptor NKG2C show robust proliferation, persistence, recall response, and augmented cytotoxicity in response to cytomegalovirus infection, demonstrating the existence of memory NK cells in humans[9,20,21].

Previous studies have demonstrated the role of NK receptors, cytokines, and transcription factors in the differentiation of memory NK cells in mice[22–29]. However, these studies have primarily focused on the quantity of memory NK cells rather than their qualitative traits. Recent studies have revealed unique transcriptional and epigenetic landscapes in memory NK cells[30,31], suggesting that the qualitative and quantitative traits of memory NK cells are intricately regulated by transcriptional regulation, epigenetic modification, and signaling through NK receptors and cytokine receptors. Furthermore, a limited number of studies have demonstrated negative regulation of memory NK cell differentiation[23,29]. However, the molecular mechanisms that qualitatively and quantitatively regulate the differentiation of memory NK cells are yet to be fully understood.

In this work, we identify Themis2 as a critical negative regulator of the differentiation of memory NK cells and investigate the role of Themis2 in NK cell memory formation and function. We show that Themis2 in the cytoplasm suppresses Ly49H signaling by attenuating ZAP70/Syk phosphorylation. Moreover, Themis2 translocates into the nucleus and promotes Zfp740-mediated repression to regulate the persistence of memory NK cells. Our study shows that Themis2 quantitatively and qualitatively regulates NK cell memory formation.

## Results

### Themis2 negatively regulates NK cell memory formation

To identify critical regulators for memory NK cell differentiation, we used NKp46-CreERT2 Tg mice with Rosa26-yellow fluorescent protein (YFP)/YFP alleles (hereinafter, NK-CreERT2 mice)[19]. NKp46+ cells in NK-CreERT2 mice express YFP upon CreERT2-mediated excision of the loxP-floxed STOP sequence in the Rosa26 loci after tamoxifen administration. Thus, NK-CreERT2 mice allowed us to track MCMV-primed NK cells, irrespective of NK cell subsets expressing or not expressing Ly49H, as YFP+ NK cells after tamoxifen administration on days 0, 1, and 2, following MCMV infection on day 0 (Fig. 1a). Considering that the vast majority of NKp46+NK1.1+TCRβ- lymphocytes in the spleen are conventional NK cells[32,33], NKp46+ type 1 and type 3 innate lymphoid cells in the spleen would be negligible in this study on naïve and MCMV-primed NK cells in the spleen. On day 10 post-infection (pi), we

purified YFP+ cells from the spleen, which included MCMV-primed effector Ly49H+KLRG1high or Ly49H-KLRG1high NK cells. On day 28 pi, we further purified MCMV-primed long-lived memory NK cells with the stringent phenotypic definition of Ly49H+KLRG1highLy6C+DNAM-1- to low and cytokine-activated Ly49H-KLRG1high NK cells (Fig. 1a and Supplementary Fig. 1). We then subjected these cell subsets to RNA-seq and obtained 51,826 unique transcripts (feature IDs), 44 of which had expressions that were upregulated in effector Ly49H+ NK cells and maintained in memory NK cells, as compared with their expressions in naïve Ly49H+ or Ly49H- NK cells, effector Ly49H- NK cells, or cytokine-activated NK cells (Fig. 1b). Because memory NK cells display transcriptomic and epigenetic profiles distinct from those of naïve and effector NK cells[31], we further narrowed down the candidates to three genes − Themis2, Trp73, and Trib1 − by their nuclear localization signal (NLS) or subcellular location by using the UniProt knowledgebase (https://www.uniprot.org) (Fig. 1b). Among three genes, we chose Themis2 as the focus of the present study because we confirmed expression in naïve NK cells at the protein level (Supplementary Fig. 2a).

A Themis family member Themis2 is known as a signaling scaffold protein that regulates activation signaling through Toll-like receptor 4 and B cell receptor in macrophages and B cells, respectively[34–36]. Themis2 mRNA expression was upregulated in effector Ly49H+ NK cells after MCMV infection, and this upregulation was maintained in the subsequent memory NK cells, whereas its expression in Ly49H- NK cells remained unchanged after MCMV infection (Supplementary Fig. 2b), which is consistent with the expression kinetics identified in an RNA-seq analysis (Supplementary Fig. 2c). Consistent with the kinetics of Themis2 mRNA, Themis2 protein expression was increased in effector Ly49H+ NK cells after MCMV infection and maintained in memory NK cells (Supplementary Fig. 2d). A functional protein association network analysis using the STRING database v10.5 (https://string-db.org) demonstrated enrichment of mouse Themis2 and human THEMIS2 in the KEGG Pathway "Natural killer cell-mediated cytotoxicity" pathways (mmu04650 and hsa04650, respectively) (Supplementary Fig. 2e), indicating that Themis2 may be involved in NK cell function. NK cells in naïve wild-type (WT) and Themis2-deficient (Themis2−/−) mice[34] did not show significant differences in the percentage and development of NK cells (Supplementary Fig. 2f, g).

Activation signaling through Ly49H is required for MCMV-primed expansion and differentiation of Ly49H+ NK cells into memory NK cells[5,19]. We investigated the role of Themis2 in activation signaling through Ly49H. Themis2−/− NK cells showed greater degranulation and phosphorylation of ZAP70 and Syk, both of which are downstream molecules of the Ly49H-DAP12 adaptor complex[16,17], after stimulation with the anti-Ly49H mAb, as compared with that in WT NK cells (Supplementary Fig. 3a, b). Together, these results indicate that Themis2 inhibits NK cell activation via Ly49H. Furthermore, Themis2−/− NK cells showed increased degranulation after stimulation of NKG2D, NKp46, and NK1.1 (NKR-P1C) (Supplementary Fig. 3c).

To further investigate mechanistic details of Themis2-mediated inhibition of Ly49H signaling, we established a transfectant of a human NK cell line NKL, which expresses the endogenous DAP12 and ZAP70, but not SYK, expressing transfected Ly49H and FLAG-tagged THEMIS2. Biochemical analysis demonstrated that THEMIS2 was coimmunoprecipitated with DAP12 and ZAP70 after stimulation with anti-Ly49H mAb when we used DTSSP for chemical crosslinking cytosolic proteins (Supplementary Fig. 3d, e). However, the coimmunoprecipitation was not detected when we did not use DTSSP (Supplementary Fig. 3d, e), suggesting that the Ly49H-mediated signal induces a protein complex formation consisting of THEMIS2, DAP12, and ZAP70 with a low affinity.

To examine the role of Themis2 in NK cells in vivo, WT and Themis2−/− mice were infected with MCMV. Themis2−/− Ly49H+ NK cells displayed a more activated phenotype than did WT Ly49H+ NK cells on day 1.5 pi, as demonstrated by increased an activation marker CD69

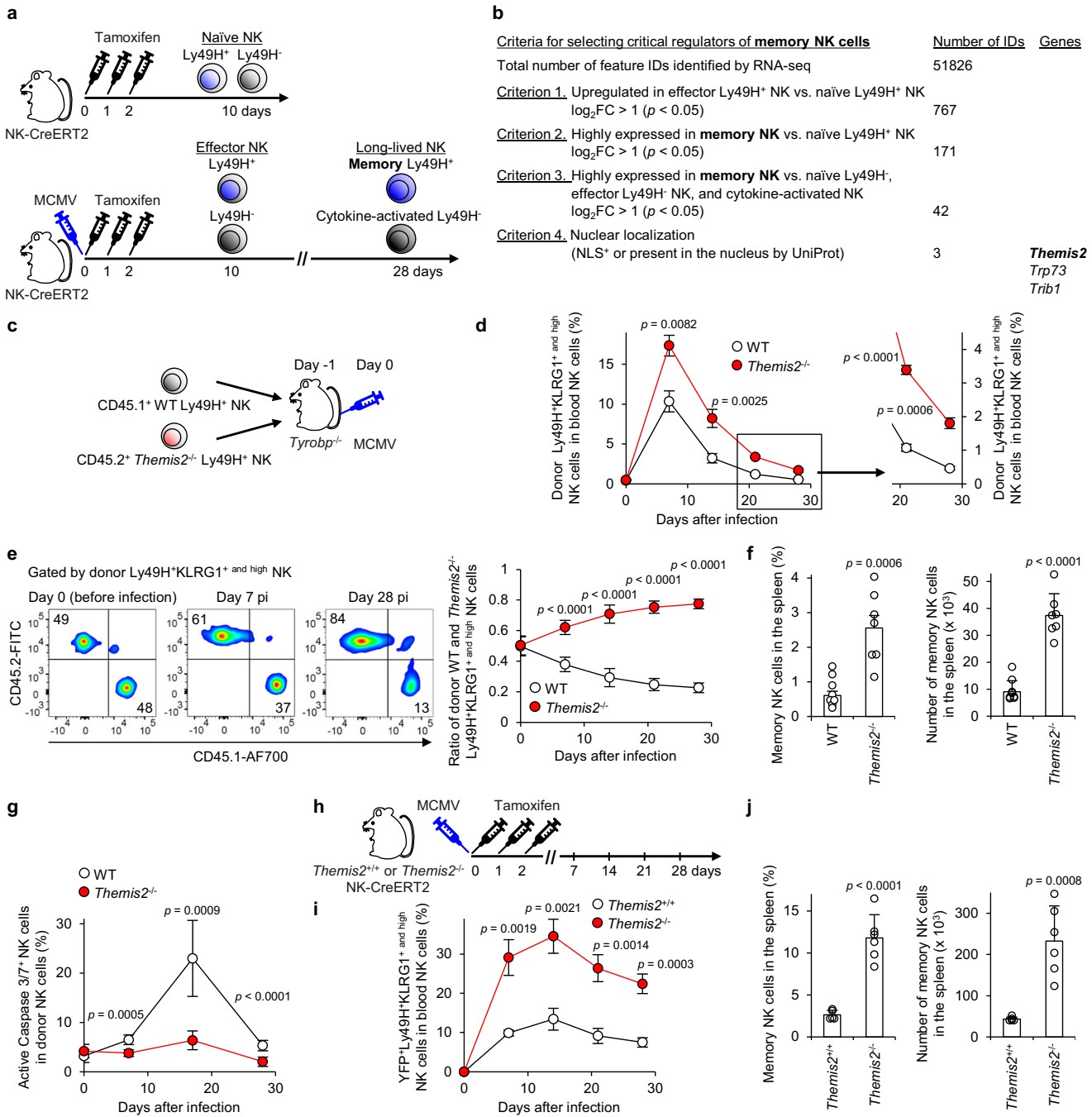

**Fig. 1 | Themis2 negatively regulates NK cell memory formation. a** Schematic representation of the approach used to obtain the natural killer (NK) cell subsets that were used to identify critical regulators of memory NK cell differentiation by RNA-seq (*n* = 3 in each group). **b** Criteria used to select critical regulators of the differentiation of memory NK cells. **c** Schematic representation of the evaluation of NK cell differentiation by co-transfer of WT and *Themis2*⁻/⁻ Ly49H⁺ NK cells into *Tyrobp*⁻/⁻ mice and mouse cytomegalovirus (MCMV) infection. **d** Percentages of donor WT and *Themis2*⁻/⁻ Ly49H⁺KLRG1⁺ and high NK cells in the blood during MCMV infection. Data are pooled from 2 experiments (*n* = 8 mice in each group). **e** Ratio of donor WT and *Themis2*⁻/⁻ Ly49H⁺KLRG1⁺ and high NK cells in the blood during MCMV infection. Flow cytometry plots are representative of 2 experiments (*n* = 4 mice in each group). Data are pooled from 2 experiments (*n* = 8 mice in each group). **f** Percentages and numbers of donor memory WT and *Themis2*⁻/⁻ NK cells in the spleen on day 28 pi. Data are pooled from 2 experiments (*n* = 8 mice (WT)

and 7 mice (*Themis2*⁻/⁻)). **g** Expression of active Caspase 3/7 in donor WT and *Themis2*⁻/⁻ Ly49H⁺KLRG1⁺ and high NK cells in the spleen during MCMV infection. Data are pooled from 2 experiments (*n* = 3 mice (day 0), 6 mice (days 7 and 17 pi), and 8 mice (day 28 pi)). **h** Schematic representation of the evaluation of NK cell differentiation by using *Themis2*⁺/⁺ and *Themis2*⁻/⁻ NK-CreERT2 mice, MCMV infection, and tamoxifen administration. **i** Percentages of *Themis2*⁺/⁺ and *Themis2*⁻/⁻ yellow fluorescent protein (YFP)⁺Ly49H⁺KLRG1⁺ and high NK cells in the blood during MCMV infection. Data are pooled from 2 experiments (*n* = 6 mice in each group). **j** Percentages and number of memory *Themis2*⁺/⁺ and *Themis2*⁻/⁻ NK cells in the spleen on day 28 pi. Data are pooled from 2 experiments (*n* = 5 mice (*Themis2*⁺/⁺) and 6 mice (*Themis2*⁻/⁻)). Statistical analysis was performed using one-way ANOVA (**d, e, g, i**) and two-sided Student's *t* test (**f, j**). Data are presented as mean values SD (**d**–**g, i, j**). Source data are provided as a Source Data file.

(Supplementary Fig. 3f). Furthermore, *Themis2*[-/-] Ly49H[+] NK cells produced a larger amount of IFN-γ than did WT Ly49H[+] NK cells on day 1.5 pi (Supplementary Fig. 3g), consistent with enhanced effector functions and ZAP70/Syk phosphorylation of *Themis2*[-/-] Ly49H[+] NK cells in vitro (Supplementary Fig. 3a, b).

To determine whether Themis2 impacts the differentiation of memory NK cells, CD45.1[+] WT Ly49H[+] NK cells and CD45.2[+] *Themis2*[-/-] Ly49H[+] NK cells were co-transferred at a ratio of 1:1 into syngeneic DAP12-deficient (*Tyrobp*[-/-]) mice lacking functional expression of Ly49H (Fig. 1c). The recipient mice were then infected with MCMV. After infection, only donor Ly49H[+] NK cells were activated in an MCMV antigen (m157)-specific manner. Expressions of the activation and differentiation markers KLRG1 and Ly6C were upregulated to the same degree on the donor WT and *Themis2*[-/-] Ly49H[+] NK cells on day 7 pi (Supplementary Fig. 4a), which is consistent with previous reports showing that MCMV-primed Ly49H[+] NK cells show upregulation of and maintain KLRG1 and Ly6C expressions throughout their differentiation into memory NK cells[19,22]. However, MCMV-primed *Themis2*[-/-] Ly49H[+]KLRG1[high] NK cells showed more robust proliferation as effector NK cells on day 7 pi, compared with WT Ly49H[+]KLRG1[high] NK cells (Fig. 1d and Supplementary Fig. 4b). Furthermore, the number and proportion of *Themis2*[-/-] Ly49H[+]KLRG1[high] NK cells were greater than those of WT Ly49H[+]KLRG1[high] NK cells in the contraction and memory phases until day 28 pi (Fig. 1d, e and Supplementary Fig. 4b). These long-lived WT and *Themis2*[-/-] Ly49H[+]KLRG1[high] NK cells on day 28 pi displayed an equivalent intact memory NK cell immunophenotype (Ly49H[+]KLRG1[high]Ly6C[+]DNAM-1[- to low]CD11b[+]CD27[-])[19,22] (Supplementary Fig. 4c). However, the number and proportion of memory *Themis2*[-/-] NK cells (stringently defined as Ly49H[+]KLRG1[high]Ly6C[+]DNAM-1[- to low]CD11b[+]CD27[-] NK cells) were larger than those of memory WT NK cells in the spleen on day 28 pi (Fig. 1f). These results indicate that Themis2 impairs control of MCMV infection and inhibits memory NK cell differentiation.

To clarify the effects of Themis2 on cell division, survival, and apoptosis of Ly49H[+] NK cells during MCMV infection, we examined the expression of Ki-67 (a cell cycle marker), Bcl-2 (a survival factor), and Bim and active Caspase 3/7 (apoptotic factors) in WT and *Themis2*[-/-] Ly49H[+] NK cells during MCMV infection. After the peak of NK cell response, the proportion of active Caspase 3/7[+] Ly49H[+] NK cells was smaller in *Themis2*[-/-] mice than in WT mice on days 7, 17, and 28 post-infection (Fig. 1g). By contrast, Themis2 deficiency had little impact on the frequency of Ki-67[+] NK cells and expression of Bim and Bcl-2 during MCMV infection (Supplementary Fig. 4d–f). These results indicate that Themis2 negatively regulates Caspase 3/7-mediated apoptosis in the contraction and memory phases.

To confirm these findings, we used *Themis2*[+/+] and *Themis2*[-/-] NK-CreERT2 mice and tracked the fate of MCMV-primed YFP[+]Ly49H[+]KLRG1[+] NK cells after tamoxifen administration following MCMV infection (Fig. 1h). We observed a higher percentage of MCMV-primed YFP[+] NK cells in the *Themis2*[-/-] NK-CreERT2 mice than in the *Themis2*[+/+] NK-CreERT2 mice on day 7 pi (Supplementary Fig. 5a). Although MCMV-primed *Themis2*[+/+] and *Themis2*[-/-] YFP[+]Ly49H[+] NK cells showed equivalent expression of KLRG1 and Ly6C on day 7 pi (Supplementary Fig. 5a), the number and proportion of *Themis2*[-/-] YFP[+]Ly49H[+]KLRG1[high] NK cells were larger than those of *Themis2*[+/+] YFP[+]Ly49H[+]KLRG1[high] NK cells in the expansion, contraction, and memory phases during MCMV infection (Fig. 1i and Supplementary Fig. 5b). MCMV-primed long-lived *Themis2*[+/+] and *Themis2*[-/-] YFP[+]Ly49H[+]KLRG1[high] NK cells on day 28 pi displayed a memory immunophenotype (defined as YFP[+]Ly49H[+]KLRG1[high]Ly6C[+]DNAM-1[- to low] NK cells) (Supplementary Fig. 5c). However, as shown in Fig. 1f, the number and proportion of memory *Themis2*[-/-] NK cells were larger than those of memory *Themis2*[+/+] NK cells in the spleen on day 28 pi (Fig. 1j). Together, these findings demonstrate that Themis2 quantitatively inhibits NK cell memory formation during MCMV infection.

## Themis2 negatively regulates the memory NK cell function

To examine whether Themis2 regulates the effector function of memory NK cells, naïve WT and *Themis2*[-/-] Ly49H[+] NK cells and memory WT and *Themis2*[-/-] NK cells derived from the spleen of recipient *Tyrobp*[-/-] mice on 28 pi as shown in Fig. 1c were co-cultured with m157-expressing RMA cells. Memory *Themis2*[-/-] NK cells showed greater degranulation and IFN-γ production after co-culture with m157-expressing RMA cells compared with memory WT NK cells and naïve *Themis2*[-/-] Ly49H[+] NK cells (Fig. 2a). To examine secondary expansion of these memory NK cells, *Tyrobp*[-/-] mice received memory WT and *Themis2*[-/-] NK cells at a ratio of one to one and were infected with MCMV. Memory *Themis2*[-/-] NK cells showed a more robust secondary expansion than did memory WT NK cells (Fig. 2b).

To analyze the function of memory *Themis2*[+/+] and *Themis2*[-/-] NK cells in the eradication of MCMV, we isolated naïve or memory Ly49H[+] NK cells from uninfected and MCMV-infected *Themis2*[+/+] and *Themis2*[-/-] NK-CreERT2 mice on day 28 pi, transferred these NK cells separately into naïve *Tyrobp*[-/-] mice, and then challenged these mice with MCMV. The memory *Themis2*[-/-] NK cells conferred greater improved protection from MCMV infection to the recipient mice than did memory *Themis2*[+/+] NK cells and naïve *Themis2*[+/+] and *Themis2*[-/-] NK cells (Fig. 2c). Together, these results indicate that Themis2 negatively regulates the quality, as well as quantity, of memory NK cells.

## Themis2 is translocated into the nucleus of NK cells

Themis2 was predicted to have a bipartite NLS. Since *Themis2*[-/-] NK cells showed enhanced memory formation after MCMV infection, it was likely that the activation signal via Ly49H induces translocation of Themis2 in the nucleus. To address the issue, we expressed GFP-fused THEMIS2 in a human NK cell line NKL expressing Ly49H. Immunofluorescence microscopy analysis showed that THEMIS2 was localized predominantly in the cytoplasm; however, stimulation with an anti-Ly49H mAb induced translocation of THEMIS2 into the nucleus, with concentrations in the nucleus peaking at 24 to 48 h after stimulation (Fig. 3a, b). To examine the subcellular localization of Themis2 in primary mouse NK cells, naïve Ly49H[+] NK cells, effector Ly49H[+] NK cells, and memory NK cells were purified from the spleen of naïve and MCMV-infected and tamoxifen-administered *Themis2*[+/+] NK-CreERT2 mice. Whereas Themis2 was predominantly detected in the cytoplasm of naïve Ly49H[+] NK cells, an equivalent amount of Themis2 was observed in the nucleus of effector Ly49H[+] NK cells and memory NK cells (Fig. 3c, d). Taken together, these results demonstrate that activation signal via Ly49H induces the nuclear translocation of Themis2 in NK cells and suggest that Themis2 is involved in transcriptional or epigenetic events for the differentiation into effector and memory NK cells after MCMV infection.

## Themis2 regulates transcriptional and epigenetic changes

To analyze the role of Themis2 in the nucleus, we performed RNA-seq and chromatin immunoprecipitation sequencing (ChIP-seq) of trimethylation at the 4[th] lysine residue of the histone H3 (H3K4me3) and acetylation at the 27[th] lysine residue of histone H3 (H3K27ac), epigenetic markers of active promoters and regulatory regions correlated with active transcription, respectively, by using naïve and memory NK cells from naive and MCMV-infected *Themis2*[+/+] and *Themis2*[-/-] NK-CreERT2 mice (Fig. 4a). A gene ontology (GO) analysis of differentially expressed genes (DEG) between memory *Themis2*[+/+] and *Themis2*[-/-] NK cells (hereinafter, memory DEGs, Supplementary Fig. 6a−c) yielded enriched signatures associated with the apoptosis, transcription, and cell cycle (Fig. 4b and Supplementary Fig. 6d). A gene set enrichment analysis (GSEA) revealed that memory *Themis2*[+/+] NK cells showed enrichment of gene sets found in activated and effector T cells, whereas memory *Themis2*[-/-] NK cells showed those found in memory T cells (Supplementary Fig. 6e), suggesting that the loss of Themis2 promotes differentiation into memory NK cells by changing the

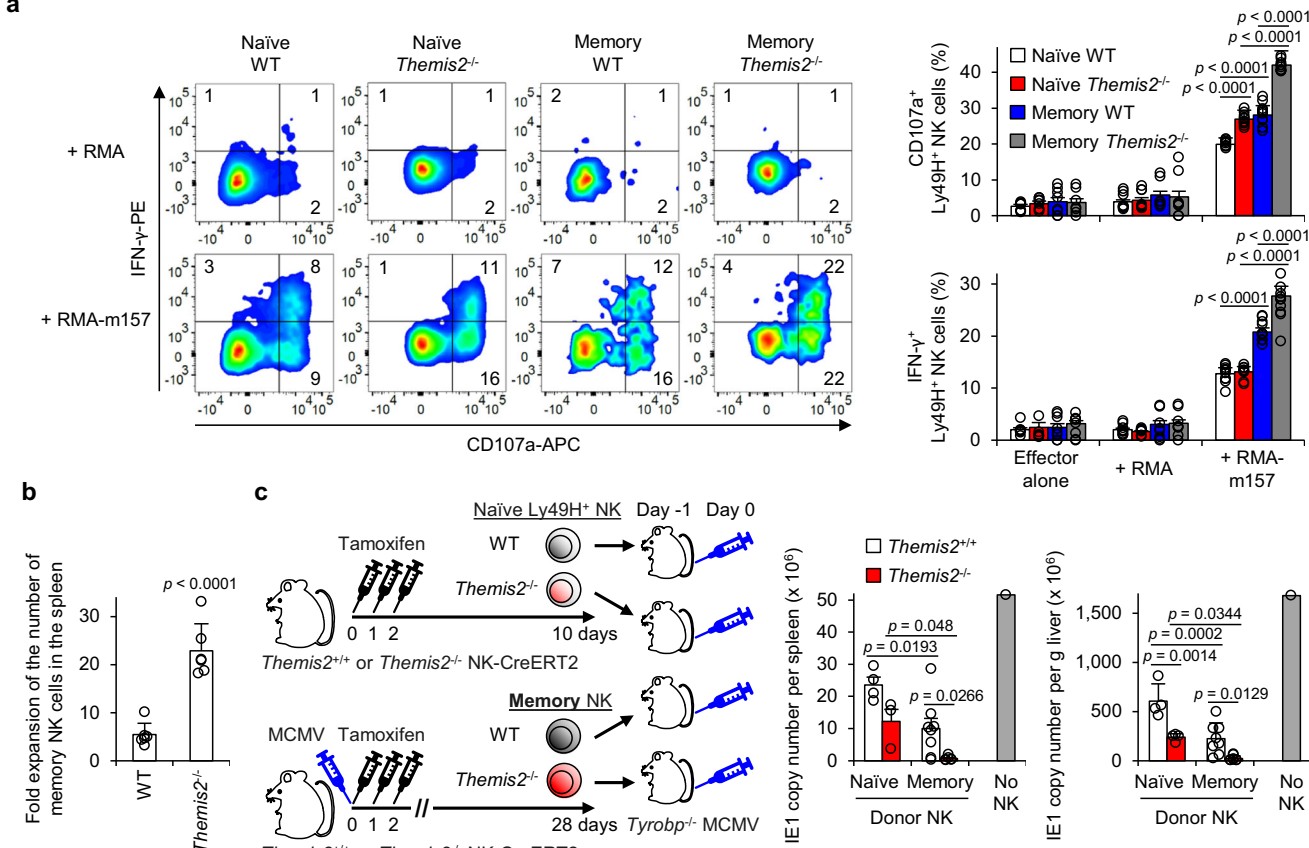

**Fig. 2 | Themis2 negatively regulates the memory NK cell function.**
**a** Degranulation and interferon-γ (IFN-γ) production of naïve and memory Ly49H⁺ WT and *Themis2⁻/⁻* Ly49H⁺ natural killer (NK) cells after co-culture with m157-expressing RMA cells. Flow cytometry plots are representative of 3 experiments (*n* = 3 wells ( + RMA) and 5 wells (+RMA-m157)). Data are pooled from 3 experiments (*n* = 8 wells (Effector alone; Naïve WT, Naïve *Themis2⁻/⁻*, Memory WT), 7 wells (Effector alone; Memory *Themis2⁻/⁻*), 9 wells ( + RMA; Naïve WT, Naïve *Themis2⁻/⁻*), 7 wells ( + RMA; Memory WT), 8 wells ( + RMA; Memory *Themis2⁻/⁻*), 10 wells ( + RMA-m157; Naïve WT, Naïve *Themis2⁻/⁻*), and 11 wells ( + RMA-m157; Memory WT, Memory *Themis2⁻/⁻*)). **b** Secondary expansion of memory WT and *Themis2⁻/⁻* NK cells in recipient *Tyrobp⁻/⁻* mice after mouse cytomegalovirus

(MCMV) infection. Secondary expansion of memory WT and *Themis2⁻/⁻* NK cells on day 7 pi is represented as the fold expansion of the number of memory NK cells in infected *Tyrobp⁻/⁻* mice relative to that in uninfected *Tyrobp⁻/⁻* mice. Data are pooled from 2 experiments (*n* = 6 mice in each group). **c** Host protection by naïve and memory NK cells. Naïve and memory *Themis2⁺/⁺* and *Themis2⁻/⁻* Ly49H⁺ NK cells were transferred separately into recipient *Tyrobp⁻/⁻* mice and then infected with MCMV. The copy number of MCMV IE1 gene in the spleen and liver was quantified on day 3 pi. Data are pooled from 2 experiments (*n* = 4 mice (Naive), 8 mice (Memory), and 1 mouse (No NK)). Statistical analysis was performed using one-way ANOVA. Data are presented as mean values ±SD (**a**–**c**). Source data are provided as a Source Data file.

transcriptional profile. Moreover, memory DEG promoters showed significant enrichment of binding motifs for transcription factors (Fig. 4c and Supplementary Fig. 6f). These findings raised the possibility that Themis2 inhibits the differentiation of memory NK cells by participating in transcriptional regulation after nuclear translocation.

*Themis2⁺/⁺* and *Themis2⁻/⁻* NK cells did not show any obvious differences in their epigenome-wide H3K4me3 and H3K27ac landscapes, irrespective of whether they were naïve and memory NK cells (Supplementary Fig. 7a). However, pairwise intersection heatmaps of the H3K4me3 and H3K27ac regions in naïve and memory NK cells revealed that memory *Themis2⁺/⁺* and *Themis2⁻/⁻* NK cells had distinct epigenetic profiles, particularly in the H3K4me3 region, as indicated by the observed negative correlation (Fig. 4d). To examine the function of the differentially accessible regions (DAR) between memory *Themis2⁺/⁺* and *Themis2⁻/⁻* NK cells, the H3K4me3 DARs uniquely present in either memory WT or *Themis2⁻/⁻* NK cells were extracted (Supplementary Fig. 7b), and their single nearest genes (hereafter, memory H3K4me3 DAR genes) were identified (Supplementary Fig. 7c). GO analysis of the memory H3K4me3 DAR genes revealed signatures associated with the cell cycle, apoptosis, and transcription (Fig. 4e). Memory H3K4me3 DAR gene promoters showed significant enrichment of binding motifs

for members of the KLF, C2H2 zinc finger, and the ETS families (Fig. 4f and Supplementary Fig. 7d). These results overlapped with those of the GO analysis and motif analysis of memory DEGs (Fig. 4c and Supplementary Fig. 6f).

To examine the non-promoter H3K27ac-marked regions, H3K27ac DARs uniquely present in either memory *Themis2⁺/⁺* or *Themis2⁻/⁻* NK cells were extracted; H3K27ac DARs overlapping with CCCTC-binding factor (CTCF)-bound distal and proximal enhancer-like signatures (ELS) available in ENCODE (https://www.encodeproject.org/) were further extracted to highlight non-promoter H3K27ac regions; and then their single nearest genes (hereinafter, memory H3K27ac DAR distal or proximal ELS genes) were identified (Supplementary Fig. 7e). Unlike memory H3K4me3 DAR genes, memory H3K27ac DAR distal ELS genes showed enriched signatures associated with the activation and differentiation of lymphocytes and leukocytes (Supplementary Fig. 7f). In contrast, memory H3K27ac DAR proximal ELS genes showed enriched signatures associated with the cell cycle (Supplementary Fig. 7g).

We assessed the correlation of H3K4me3 and H3K27ac landscapes in naïve and memory *Themis2⁺/⁺* or *Themis2⁻/⁻* NK cells by using the unified atlas of these H3K4me3 and H3K27ac regions. The pairwise

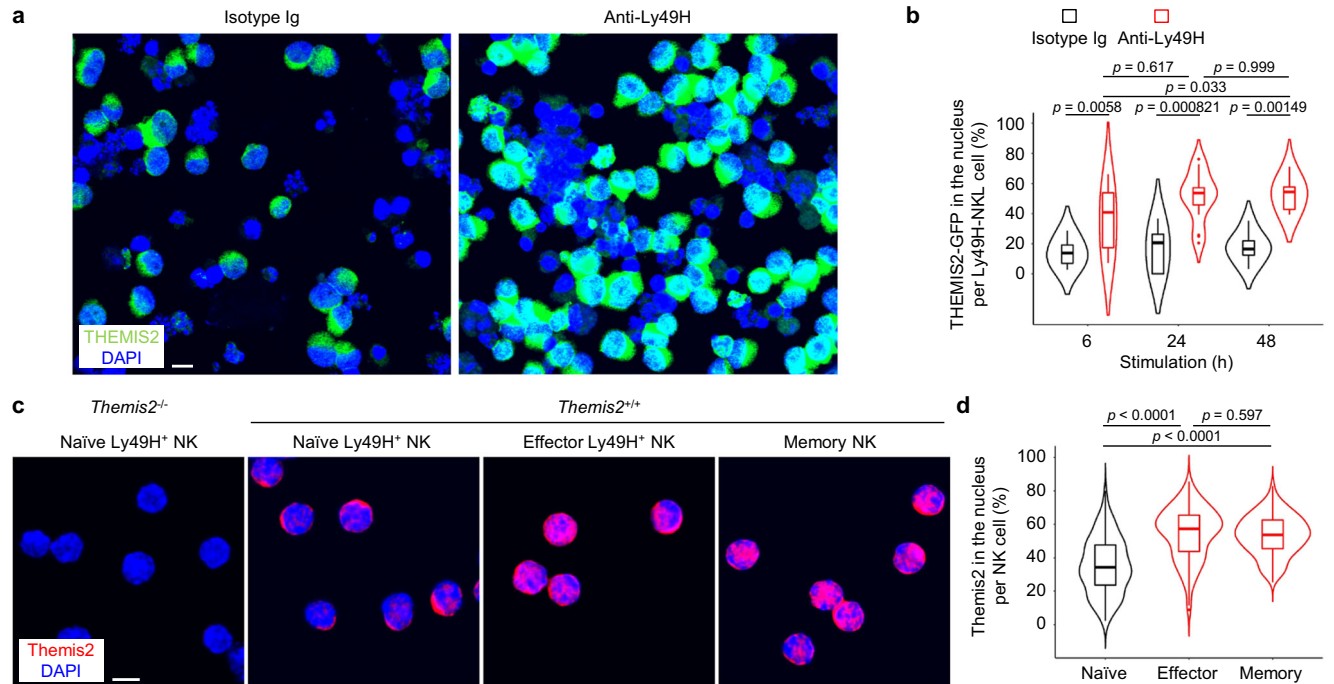

**Fig. 3 | Themis2 is translocated into the nucleus of NK cells. a** Subcellular localization of THEMIS2 in a human natural killer (NK) cell line NKL-Ly49H cells expressing green fluorescent protein (GFP)-fused THEMIS2 after stimulation with anti-Ly49H mAb for 24 h. Bar, 10 μm. Images are representative of 5 experiments ($n = 4$ images (Isotype Ig) and 9 images (Anti-Ly49H)). **b** Percentages of THEMIS2 in the nucleus per NKL-Ly49H cell after stimulation with anti-Ly49H mAb. Data are pooled from 2 experiments ($n = 9$ cells), 9 cells, 7 cells (6, 24, 48 h of Isotype Ig, respectively) and 16 cells, 19 cells, 11 cells (6, 24, 48 h of Anti-Ly49H, respectively). **c** Subcellular localization of Themis2 in primary mouse NK cells. Naïve yellow fluorescent protein (YFP)$^+$Ly49H$^+$KLRG1$^+$ NK cells, effector YFP$^+$Ly49H$^+$KLRG1$^{high}$ NK cells, and memory YFP$^+$Ly49H$^+$KLRG1$^{high}$ NK cells were purified from the spleen of naïve and mouse cytomegalovirus (MCMV)-infected and tamoxifen-administered *Themis2*$^{+/+}$ and *Themis2*$^{-/-}$ NK-CreERT2 mice and subcellular localization of Themis2 was assessed by immunofluorescence staining. Bar, 5 μm. Images are representative of 2 experiments ($n = 4$ images (*Themis2*$^{-/-}$ Naïve Ly49H$^+$ NK), 6 images (*Themis2*$^{+/+}$ Naïve Ly49H$^+$ NK), 8 images (*Themis2*$^{+/+}$ Effector Ly49H$^+$ NK and *Themis2*$^{+/+}$ Memory NK)). **d** Percentages of Themis2 in the nucleus per cell for each the NK cell subsets. Data are pooled from 2 experiments ($n = 66$ cells (Naïve), 58 cells (Effector), 82 cells (Memory)). Statistical analysis was performed using one-way ANOVA. Violin plots show median, minimum, maximum, 25% and 75% percentiles, outliers, and kernel density estimation. Source data are provided as a Source Data file.

intersection heatmap revealed that Themis2 has little impact on the overall correlations in H3K4me3 and H3K27ac regions in all NK cell subsets (Supplementary Fig. 7h). We also examined the correlation of gene expression of memory DEGs and signals of memory H3K4me3 DARs. The scatter plots suggested the possibility that Themis2 might be involved in gene expression and promoter accessibility of transcription-associated genes (Supplementary Fig. 7i).

Together, these findings support the hypothesis that Themis2 negatively regulates NK cell memory formation through the transcriptional regulation of apoptosis-associated genes in the nucleus, in addition to the negative regulation of activation signaling via Ly49H in the cytoplasm.

**Themis2 promotes Zfp740-mediated transcriptional repression**
To identify Themis2-binding nuclear proteins, the nuclear proteins of MCMV-primed effector Ly49H$^+$ NK cells from the spleen of MCMV-infected and tamoxifen-administered *Themis2*$^{-/-}$ NK-CreERT2 mice were incubated with recombinant FLAG-tagged Themis2 protein (Themis2-FLAG); the Themis2-bound proteins were then immunoprecipitated with an anti-FLAG mAb (Supplementary Fig. 8a–c and Fig. 5a), and analyzed by matrix-assisted laser desorption-ionization time-of-flight (MALDI-TOF) mass spectrometry. A MASCOT search identified nine transcription factors and one epigenetic enzyme expressed in NK cells (Supplementary Fig. 8d), implying that Themis2 preferentially interacts with transcription factors in the nucleus of NK cells.

To identify the key transcription factor for memory NK cell differentiation, we analyzed the lists of transcription factors predicted by motif analyses of memory DEG promoters and memory H3K4me3 DAR

gene promoters (Fig. 4c, f, and Supplementary Fig. 6f, 7d) and compared those lists with a list of Themis2-bound transcription factors detected by MADLI-TOF mass spectrometry (Fig. 5b). Consequently, Zfp740, a zinc finger protein with unknown function, was found to be common among the three lists of transcription factors (Fig. 5b). Mouse Zfp740 (UniProt Q6NZQ6) is a C2H2 zinc finger transcription factor with three zinc finger domains and a low complexity disordered region. Mouse Zfp740 and human ZNF740 (UniProt Q8NDX6) have high homology (Blastp E-value, $3 \times 10^{-114}$), and their C-terminus amino acid sequences containing zinc finger domains are identical. Although no studies have reported the role of mouse Zfp740, Ingenuity Pathway Analysis revealed that human ZNF740 has an RNA polymerase II-specific transcription factor with DNA-binding activity and that it interacts with Polycomb-group proteins BMI1 (aka PCGF4) and RNF2 (aka RING1B) (Supplementary Fig. 8e), both of which are core components of Polycomb repressive complex 1[37]. These results suggest that Zfp740 might be involved in transcriptional repression by interacting with transcriptional repressor complexes. Zfp740 was located in the nucleus in 293 T cells expressing MYC-tagged Zfp740 protein (Zfp740-MYC), as determined by immunoblotting analysis with anti-MYC mAb (Supplementary Fig. 8f). Moreover, nuclear Themis2 protein was physically associated with the Zfp740 protein in 293 T cells co-expressing Zfp740-MYC and Themis2-FLAG with an additional NLS (referred to as NLS-Themis2-FLAG) for efficient nuclear localization (Fig. 5c and Supplementary Fig. 8f).

To determine the mode of action of Zfp740 as a transcription factor, we designed luciferase reporter constructs driven by a promoter harboring either the Zfp740-binding motif predicted by the

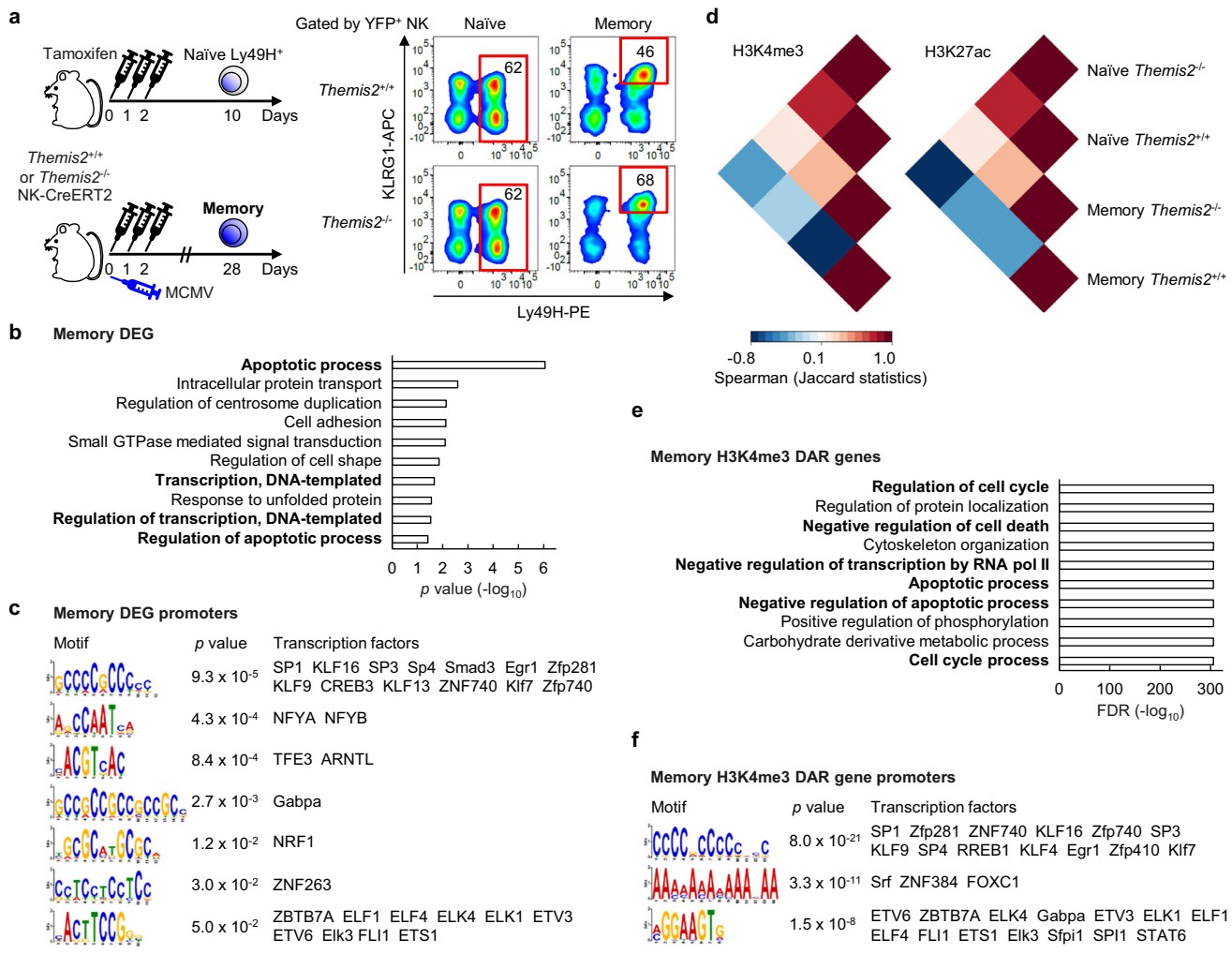

**Fig. 4 | Themis2 regulates transcriptional and epigenetic changes in memory NK cells. a** Schematic representation of the preparation of naïve and memory *Themis2*⁺/⁺ and *Themis2*⁻/⁻ natural killer (NK) cells for RNA sequencing (**b**, **c**) and chromatin immunoprecipitation sequencing of tri-methylation at the 4th lysine residue of the histone H3 (H3K4me3) regions (**d**–**f**) and acetylation at the 27th lysine residue of histone H3 (H3K27ac) regions (*n* = 3 in each group). Naïve *Themis2*⁺/⁺ and *Themis2*⁻/⁻ yellow fluorescent protein (YFP)⁺Ly49H⁺ NK cells were purified from the spleen of uninfected and tamoxifen-administered *Themis2*⁺/⁺ and *Themis2*⁻/⁻ NK-CreERT2 mice on day 10. Memory *Themis2*⁺/⁺ and *Themis2*⁻/⁻ YFP⁺Ly49H⁺KLRG1^high NK cells were purified from the spleen of mouse cytomegalovirus (MCMV)-infected and tamoxifen-administered *Themis2*⁺/⁺ and *Themis2*⁻/⁻ NK-CreERT2 mice on day 28 pi. Red boxes represent the gating of naïve and memory NK cells. **b** Gene ontology (GO) analysis of memory differentially

expressed genes (DEG). GO terms of Biological Process are shown with *p* values. **c** Motif analysis of memory DEGs. Transcription factors that can bind to these binding sequences enriched in memory DEG promoters are shown with *p* values. **d** Pairwise intersections of individual atlases of H3K4me3 and H3K27ac in naïve and memory *Themis2*⁺/⁺ and *Themis2*⁻/⁻ NK cells. Pairwise intersection heatmaps are shown with Spearman's correlation values. **e** GO analysis of memory H3K4me3 differentially accessible region-associated (DAR) genes. GO terms of Biological Process are shown with false discovery rate (FDR). **f** Motif analysis of memory H3K4me3 DAR genes. Transcription factors that can bind to these binding sequences enriched in memory H3K4me3 DAR gene promoters are shown with *p* values. The *p* value for each motif was calculated by a one-sided statistical test of MEME Tomtom (see the "Methods" section). Source data are provided as a Source Data file.

motif analyses of memory DEG promoters and memory H3K4me3 DAR gene promoters (Fig. 4c, f) or a random motif, and transiently expressed with Zfp740 in 293 T cells. Zfp740 reduced the luciferase activity in a Zfp740-binding motif-dependent manner (Fig. 5d). Next, we addressed the effect of nuclear Themis2 on the transcriptional activity of Zfp740 by co-expression with or without NLS-Themis2. Although NLS-Themis2 alone had little impact on luciferase activity, co-expression of NLS-Themis2 and Zfp740 significantly reduced it, as compared with Zfp740 alone (Fig. 5e). These results indicate that Zfp740 is a transcriptional repressor and Themis2 promotes Zfp740-mediated transcriptional repression.

## Zfp740 inhibits differentiation of memory NK cells

*Zfp740* mRNA was ubiquitously expressed in the organs of mice and similarly expressed in most immune cell subsets, including naïve,

effector, and long-lived Ly49H⁺ and Ly49H⁻ NK cell subsets (Supplementary Fig. 9a–c). We generated Zfp740-deficient (*Zfp740*⁻/⁻) mice, and these mice showed normal NK cell development (Supplementary Fig. 9d–g). To determine the role of Zfp740 in NK cell memory formation, *Tyrobp*⁻/⁻ mice received CD45.1⁺ WT Ly49H⁺ NK cells and CD45.2⁺ *Zfp740*⁻/⁻ Ly49H⁺ NK cells at a 1:1 ratio and were then infected with MCMV. The activation and differentiation markers KLRG1 and Ly6C were equivalently upregulated on the MCMV-primed donor WT and *Zfp740*⁻/⁻ Ly49H⁺ NK cells on day 7 pi (Supplementary Fig. 9h). Unlike MCMV-primed effector *Themis2*⁻/⁻ Ly49H⁺KLRG1^high NK cells (Fig. 1d), the proliferation of MCMV-primed effector *Zfp740*⁻/⁻ Ly49H⁺KLRG1^high NK cells was comparable to that of effector WT Ly49H⁺KLRG1^high NK cells on day 7 pi (Fig. 6a). However, MCMV-primed *Zfp740*⁻/⁻ KLRG1^highLy49H⁺ NK cells showed efficient persistence in the contraction phase, and the number and proportion of *Zfp740*⁻/⁻

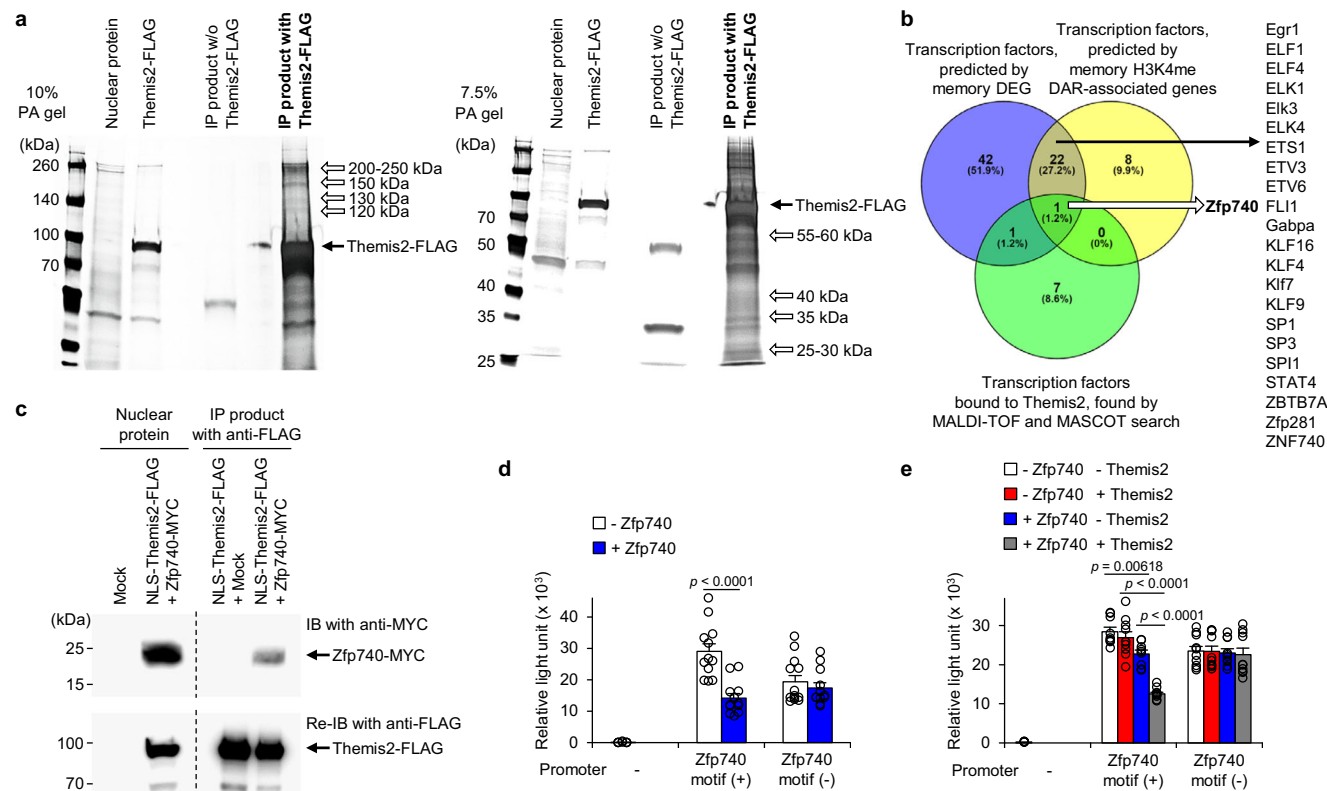

**Fig. 5 | Themis2 promotes Zfp740-mediated transcriptional repression.** Nuclear proteins of effector *Themis2⁻/⁻* natural killer (NK) cells were incubated with Themis2-FLAG, and Themis2-bound proteins were immunoprecipitated. Unique protein bands of the immunoprecipitated (IP) products were analyzed by mass spectrometry. **a** Silver staining of IP products. Proteins bands uniquely present in IP products incubated with Themis2-FLAG are shown by white arrows. These images are representative of 5 experiments. **b** Venn diagram of transcription factors predicted by motif analyses of memory differentially expressed gene (DEG) promoters, and memory tri-methylation at the 4ᵗʰ lysine residue of the histone H3 (H3K4me3) differentially accessible region-associated (DAR) gene promoters, and Themis2-bound transcription factors detected by mass spectrometry. **c** Interaction of Themis2 with Zfp740 in the nucleus. Themis2-FLAG in the nuclear proteins of 293 T co-expressing nuclear localization signal (NLS)-Themis2-FLAG and Zfp740-MYC was immunoprecipitated with anti-FLAG mAb and immunoblotted (IB) with anti-MYC mAb, followed by reblotting with anti-FLAG mAb. The image is representative of 3

experiments. **d** Luciferase reporter assay with Zfp740. Luciferase reporter constructs with Zfp740-binding or random motifs were transfected with or without Zfp740 into 293 T cells. The transcriptional activity is represented as relative light unit. Data are pooled from 4 experiments (*n* = 8 wells (no promoter), 12 wells (Zfp740 motif (+)), and 13 wells (Zfp740 motif (−))). **e** Luciferase reporter assay with Zfp740 in the presence or absence of Themis2. Luciferase reporter constructs with Zfp740-binding or random motifs were transfected with or without Zfp740 and or NLS-Themis2 into 293 T cells. The transcriptional activity is represented as relative light unit. Data are pooled from 3 experiments (*n* = 6) wells (no promoter), 9 wells (Zfp740 motif (+);−Zfp740−Themis2, + Zfp740−Themis2), and 10 wells (Zfp740 motif (+);−Zfp740 + Themis2, + Zfp740 + Themis2, and Zfp740 motif (−)). Statistical analysis was performed using two-sided Student's *t* test (**d**) and one-way ANOVA (**e**). Data are presented as mean values ±SD (**d**, **e**). Source data are provided as a Source Data file.

Ly49H⁺KLRG1^high NK cells were greater than those of WT Ly49H⁺KLRG1^high NK cells in the contraction and memory phases until day 28 pi (Fig. 6a, b and Supplementary Fig. 9i), which is consistent with the kinetics of MCMV-primed *Themis2⁻/⁻* Ly49H⁺KLRG1^high NK cells (Fig. 1d, e). These long-lived WT and *Zfp740⁻/⁻* Ly49H⁺KLRG1^high NK cells on day 28 pi displayed an equivalent memory immunophenotype (Ly49H⁺KLRG1^high Ly6C⁺DNAM-1^- to low CD11b⁺CD27⁻) (Supplementary Fig. 9j). However, a higher percentage and number of MCMV-primed memory *Zfp740⁻/⁻* NK cells (stringently defined as Ly49H⁺KLRG1^high Ly6C⁺DNAM-1^-to low CD11b⁺CD27⁻ NK cells) were present in the spleen on day 28 pi, as compared with the MCMV-primed memory WT NK cell population (Fig. 6c). These results demonstrate that Zfp740 inhibits NK cell memory formation during MCMV infection, even though Zfp740 is dispensable for the expansion of effector NK cells.

Memory *Themis2⁻/⁻* NK cells highly expressed anti-apoptosis-associated genes with GO terms related to the negative regulation of apoptosis, such as *Niban2*, *Bcl10*, *Map4k4*, and *Sirt1*[38–41], whose promoters had putative Zfp740-binding sequences, as compared with memory WT NK cells (Supplementary Fig. 9k). Consistent with these results, memory *Zfp740⁻/⁻* NK cells highly expressed the anti-

apoptosis-associated genes with promoters bearing putative Zfp740-binding sequences, compared with memory WT NK cells (Fig. 6d). These findings support the possibility that Themis2 and Zfp740 inhibit the differentiation and persistence of memory NK cells through transcriptional repression of anti-apoptosis-associated genes during MCMV infection.

To investigate whether Zfp740 impacts the effector function of NK cells, naïve and memory WT and *Zfp740⁻/⁻* Ly49H⁺ NK cells were co-cultured with m157-expressing RMA cells. Although naïve WT and *Zfp740⁻/⁻* Ly49H⁺ NK cells degranulated and produced IFN-γ against m157-expressing RMA cells to the same degree, memory *Zfp740⁻/⁻* NK cells showed more efficient degranulation and IFN-γ production against m157-expressing RMA cells compared with memory WT NK cells and naïve WT and *Zfp740⁻/⁻* Ly49H⁺ NK cells (Fig. 6e). These findings demonstrate that Zfp740 suppresses the effector function of memory, but not of naïve, NK cells. Thus, Zfp740 also negatively regulates the quantity and quality of NK cell memory formation.

## Discussion
In this study, we identified Themis2 as a critical regulator of the differentiation and function of memory NK cells during MCMV infection.

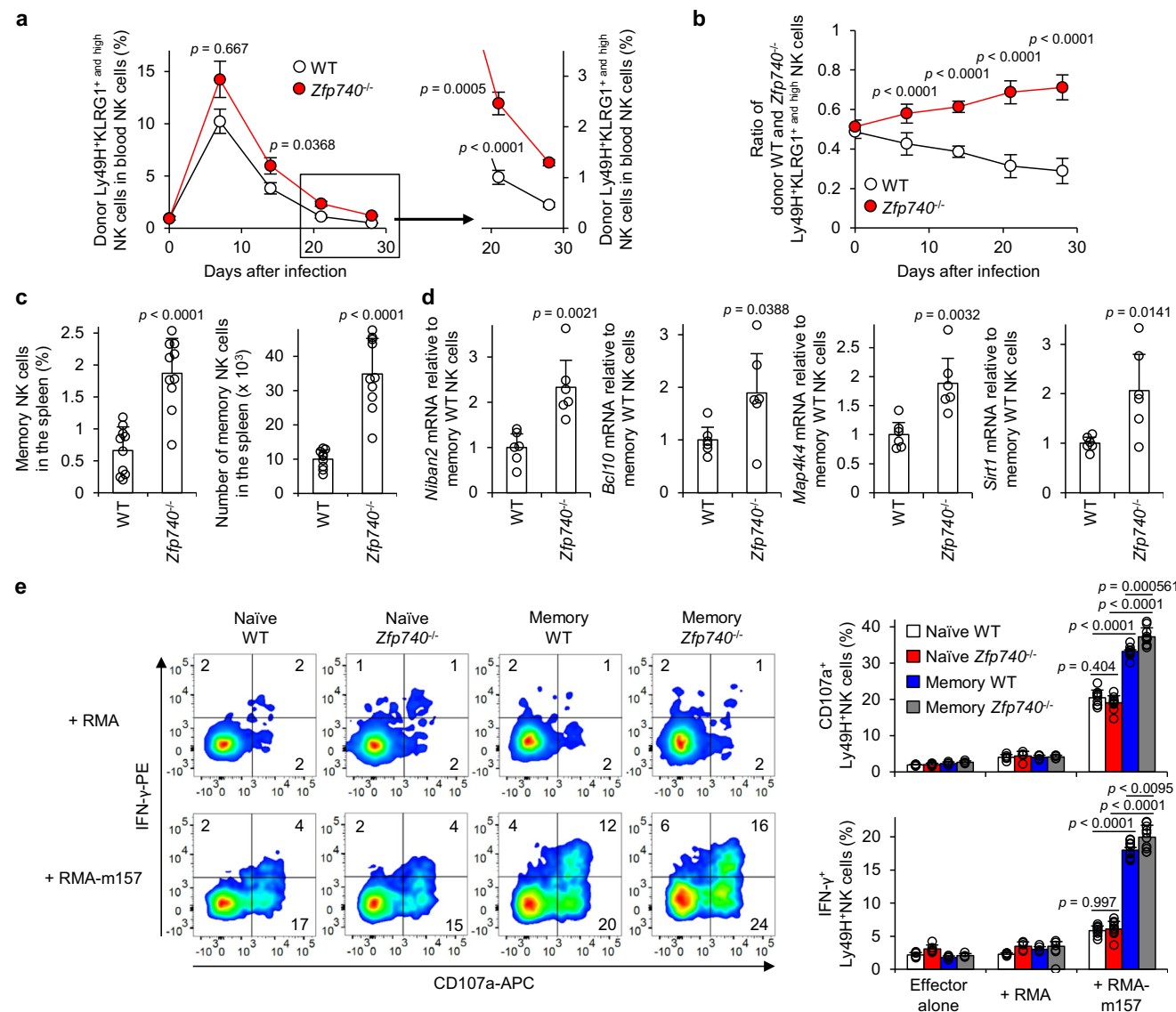

**Fig. 6 | Zfp740 inhibits differentiation and effector function of memory NK cells.** Recipient *Tyrobp*[−/−] mice received both donor CD45.1[+] WT Ly49H[+] natural killer (NK) cells and CD45.2[+] *Zfp740*[−/−] Ly49H[+] NK cells at a 1:1 ratio and then infected with mouse cytomegalovirus (MCMV). **a** Percentages of donor WT and *Zfp740*[−/−] Ly49H[+]KLRG1[+ and high] NK cells in the blood during MCMV infection. Data are pooled from 2 experiments (*n* = 10 mice in each group). **b** Ratio of donor WT and *Zfp740*[−/−] Ly49H[+]KLRG1[+ and high] NK cells in the blood during MCMV infection. Data are pooled from 2 experiments (*n* = 10 mice in each group). **c** Percentages and number of donor memory WT and *Zfp740*[−/−] NK cells in the spleen on day 28 pi. Data are pooled from 2 experiments (*n* = 10 mice in each group). **d** Expression of mRNA of anti-apoptosis-associated genes with promoters bearing putative Zfp740-binding sequences in memory *Zfp740*[−/−] NK cells relative to memory WT NK cells was analyzed by quantitative reverse transcription PCR (qRT-PCR). Data are pooled from 2 experiments (*n* = 6 mice in each group). **e** Degranulation and interferon-γ (IFN-γ) production of naïve and memory WT and *Zfp740*[−/−] Ly49H[+] NK cells after co-culture with m157-expressing RMA cells. Flow cytometry plots are representative of 2 experiments (*n* = 3 wells (+RMA) and 5 wells (+RNA-m157)). Data are pooled from 2 experiments (*n* = 6 wells (Effector alone; Naïve WT, Naïve *Zfp740*[−/−]), 4 wells (Effector alone; Memory WT, Memory *Zfp740*[−/−]), 6 wells (+RMA; Naïve WT, Memory WT), 5 wells (+RMA; Naïve *Zfp740*[−/−], Memory *Zfp740*[−/−]), 10 wells (+RMA-m157)). Statistical analysis was performed using one-way ANOVA (**a, b, e**) and two-sided Student's *t* test (**c, d**). Data are presented as mean values ±SD (**a**–**e**). Source data are provided as a Source Data file.

Themis2 inhibited activation signaling via Ly49H by attenuating the activation of ZAP70 and/or Syk, which restricted effector function and expansion of memory Ly49H[+] NK cells. Moreover, Themis2 was translocated into the nucleus in Ly49H[+] NK cells following MCMV infection. In the nucleus, Themis2 promoted Zfp740-mediated transcriptional repression of anti-apoptosis-associated genes to regulate the persistence of memory NK cells. Thus, Themis2 qualitatively and quantitatively limits the formation of NK cell memory (Supplementary Fig. 10).

There is limited literature describing the function of Themis2 in immune cells. In contrast to an inhibitory effect of Themis2 on Ly49H

signaling, previous reports have demonstrated that Themis2 promotes activation of ERK and p38, augmenting Toll-like and B cell receptor signals in macrophages and B cells, respectively[34,35]. These findings imply that Themis2 exerts an activating or inhibitory function in a stimulus- or cell-type-dependent manner. Themis1 is a well-studied Themis family protein with similar domain architectures to those of Themis2[36]. Although Themis1 is required for the activation of thymocytes through modulation of T cell receptor signaling[42,43], more recent studies have revealed that Themis1 forms a complex with SHP-1 and increases phosphatase activity through stabilization of phosphorylated SHP-1[44,45]. Considering the functional interchangeability of

Themis1 and Themis2 in thymocyte development, as well as the similarity of the activation through T cell receptor and Ly49H, both of which transmit activation signals via signaling immunoreceptor tyrosine-based activation motif-bearing adaptor molecules (CD3ζ and DAP12, respectively) and ZAP70[14,46,47], Themis2 might bind to SHP-1 and inhibit activation signaling in NK cells through increased phosphatase activity[14,48]. The phosphorylation of ZAP70/SYK, which are signaling molecules downstream of DAP12, triggers the activation of multiple signaling adaptors and enzymes, including PLCγ1/2[48]. PLCγ1/2 catalyzes the production of the second messengers diacylglycerol and inositol(1,4,5)-trisphosphate, which triggers PKC activation and an increased $Ca^{2+}$ release from the endoplasmic reticulum, respectively, and the subsequent $Ca^{2+}$-calcineurin signaling, resulting in nuclear translocation of NFAT[48]. Thus, Themis2 might promote SHP-1-mediated dephosphorylation of the protein substrates of the tyrosine kinases linked to activating NK receptors, including Ly49H, which contributes to the downregulation of the $Ca^{2+}$ influx. Notably, SHP-1 limits the proliferation of Ly49H[+] NK cells and host protection against MCMV infection[49]. Thus, Themis2 likely forms a complex with SHP-1 and presumably with DAP12 and or ZAP70 (Supplementary Fig. 3d and 3e), and attenuates Ly49H signaling, resulting in restricted effector function of naïve and memory NK cells. Furthermore, naïve Themis2[-/-] NK cells showed augmented effector functions after stimulation of NKG2D, NKp46, and NK1.1 (NKR-P1C), as well as Ly49H, suggesting that the cytoplasmic Themis2 may have an inhibitory role in activation signaling through a diverse group of activating NK receptors, irrespective of naïve or memory NK cells. Interestingly, Zfp740 also inhibited the effector function of memory NK cells. Thus, the enhanced effector function of memory Themis2[-/-] NK cells may be attributable to the loss of Themis2-mediated inhibition of activation signaling in the cytoplasm and the loss of Themis2-Zfp740 complex-mediated repression in the nucleus.

Intriguingly, Themis2 regulated Caspase 3/7-mediated apoptosis of MCMV-primed effector and memory NK cells in the contraction and memory phases, whereas Themis2 had little impact on the apoptosis in naïve NK cells. In addition, Themis2 inhibited effector functions and proliferation of naïve and memory Ly49H[+] NK cells. These results suggest that Themis2 regulates the effector function and or survival of naïve and differentiated NK cells. On the other hand, although Zfp740 regulated the survival and the effector function of memory NK cells, the effector function of naïve NK cells was not affected by Zfp740. It is presumed that the Zfp740 function is specific, particularly to regulating the expression of anti-apoptosis-associated genes in memory NK cells.

The comprehensive understanding of Themis2-bound nuclear proteins in naïve, effector, and memory NK cells would be important, because differential Themis2-bound proteins may regulate NK cell memory. However, there is a technical limitation: The quantity of nuclear protein obtained from an NK cell. Mouse NK cells per spleen yield smaller than 100 µg nuclear proteins. Further, the frequency of memory NK cells is approximately 10% in splenic NK cells in Themis2[-/-] NK-CreERT2 mice (Fig. 1j). To overcome the issue, we used effector Ly49H[+] NK cells to identify Themis2-binding transcription factors, because they are increased in number on days 7-10 pi. However, it seems technically impossible to obtain an adequate amount of nuclear proteins from memory NK cells. Although we were interested in Themis2-bound nuclear proteins in naïve, effector, and memory NK cells comprehensively, effector NK cells were realistic for the pull-down and the subsequent mass spectrometry. Future studies using other methods might clarify the roles of differential Themis2-bound nuclear proteins in controlling NK cell memory.

Transcriptional repression is achieved by the combinatorial mechanisms of action of a variety of repressor complexes consisting of transcriptional repressors, co-factors, and epigenetic regulators, and often through enzymatic histone modifications for epigenetic silencing[50]. Polycomb repressive complexes 1 and 2 (PRC1 and PRC2) constitute a conserved gene silencing system that plays pivotal roles in embryogenesis, development, and differentiation processes[37]. Zfp740 is a member of the C2H2 zinc finger transcription factors that activate or repress the expression of genes involved in development and differentiation[51]. Human ZNF740 is physically associated with BMI1 and RNF2 (Supplementary Fig. 8e)[52], both of which are core components of PRC1[37]. Given the conserved function of mouse Zfp740 and human ZNF740 with their identical zinc finger domain amino acid sequences, one possible scenario is that the Themis2-Zfp740 complex participates in PRC1-mediated transcriptional repression; the complex binds to Zfp740-binding sequences of the regulatory elements and might recruit PRC1 for stable silencing of Zfp740 target genes in memory NK cells. A recent study has reported that an ETS family transcriptional factor, Fli1, restricts the formation of memory precursor Ly6C[-] Ly49H[+] NK cells with increased protein expression of Bcl-2 at the early stage of MCMV infection[29]. In the present study, the populations of WT and Themis2[-/-] Ly6C[+] Ly49H[+] NK cells showed comparable each other on day 7 pi (Supplementary Fig. 4a, 5a), and WT and Themis2[-/-] Ly49H[+] NK cells equivalently expressed Bcl-2 during MCMV infection (Supplementary Fig. 4e), suggesting that Themis2 may not be involved in the regulation of the size of the memory precursor NK cell pool. We also identified Fli1 in our motif analyses of memory DEG promoters and memory H3K4me3 DAR gene promoters (Fig. 5b). It is possible that multiple transcriptional factors, including Zfp740 and Fli1, redundantly and independently control the quantity and quality of memory NK cells with different modes of action at the same or distinct regulatory elements. The cooperative interactions of Zfp740, Fli1, and Polycomb group proteins in forming NK cell memory have not yet been examined. Further studies are needed to elucidate the mechanistic role of Zfp740 and Themis2 in Zfp740-mediated repression, i.e., identification of Zfp740 target genes and their biological processes regulated by Zfp740 repressor complex with or without Themis2. Such investigations might reveal the core molecular identity of NK cell memory, although a current technical limitation is the lack of a chromatin immunoprecipitation-validated antibody against Zfp740.

Here, we demonstrated that Themis2 deficiency reinforces expansion, persistence, effector functions, and memory differentiation of MCMV-primed Ly49H[+] NK cells and that it confers improved host protection against MCMV infection. All these features of Themis2-deficient NK cells appear beneficial for anti-viral and anti-tumor immunity. We also found that Themis2 deficiency did not disadvantage NK cells; hence, the physiological relevance of Themis2 is still unknown. The functional properties of memory NK cells are attractive for developing NK cell-based immunotherapies for incurable infectious diseases and malignancies. Thus, further studies of the detailed molecular mechanisms of Themis2-mediated inhibition of NK cell function are needed to gain important insights into the identity of NK cell memory; such studies might provide opportunities for developing therapeutic interventions to selectively unleash NK cells to combat viral infections and cancers.

## Methods

This study was approved by the Laboratory Animal Ethics Committee of the University of Tsukuba (approval number 22-154) and approved by the Ethics Committee for Medical Sciences at the University of Tsukuba (approval number 234-2).

### Mice

WT (stock number C57BL/6JJcl) and congenic CD45.1[+] B6 (B6.SJL-Ptprc[a]Pepc[b]/BoyJ, stock number 002014) were purchased from CLEA Japan (Tokyo, Japan) and the Jackson Laboratory (Bar Harbor, Maine, U.S.A.), respectively. Mice carrying inducible Cre expressed under the control of the Ncr1 gene harboring Rosa26-YFP alleles (NK-CreERT2 mice) on the B6 background and Tyrobp[-/-] B6 mice were kindly

provided by Prof. Lewis L. Lanier (University of California, San Francisco)[19,53]. For generation of NK-CreERT2 mice, NKp46-CreERT2 Tg mice, which harbor the CreERT2 recombinase expressed under the control of the mouse *Ncr1* gene, were intercrossed with *Rosa26*-YFP/YFP B6 mice carrying a *loxP*-flanked STOP sequence followed by the YFP sequence into the *Gt(ROSA)26Sor* loci. *Themis2*[-/-] B6 mice were kindly provided by Prof. Richard J. Cornall (University of Oxford)[34]. For labeling naïve or MCMV-primed NK cells, naïve or MCMV-infected *Themis2*[+/+] and *Themis2*[-/-] NK-CreERT2 mice were treated with tamoxifen on days 0-2 pi, as described previously[19]. *Zfp740*[-/-] B6 mice were generated by using guide RNA targeting exons 3-6 encoding zinc finger domains of Zfp740 (upstream target sequence, ACCCTGCCAGG AGACCTGAC [chr15:102,207,729-102,207,748]; and downstream target sequence, CGTCAGTGCAGTTAGCACCT [chr15:102,209,336-102,209,355]), as described previously[54].

Mice between 6 and 24 weeks of age were used for experiments in a gender-matched manner. All mice were housed and maintained under the specific-pathogen-free conditions and at $23.5\,°C \pm 2.5\,°C$ ambient temperature and $52.5\% \pm 12.5\%$ humidity on a 14-h light (5 am to 7 pm): 10-h dark cycle (7 pm to 5 am). The mice were fed a MF diet (MF, Oriental Yeast, Tokyo, Japan). All strains of mice were bred separately. Mice were euthanized at experimental endpoint by inhalation of 70 to 100% $CO_2$ for 30 to 60 seconds, followed by cervical dislocation. All procedures were approved by the Laboratory Animal Ethics Committee of the University of Tsukuba (approval number 22-154) and performed at the laboratory animal resource centers of the University of Tsukuba in accordance with the guidelines of the institutional animal ethics committees.

### MCMV infection

Smith strain MCMV (murid betaherpesvirus 1) was purchased from ATCC (Manassas, Virginia, U.S.A.) and the salivary gland virus were prepared as described previously[23]. Mice were i.p. injected with $1 \times 10^5$ plaque-forming units (pfu) MCMV. *Tyrobp*[-/-] mice that had received donor Ly49H[+] NK cells were intraperitoneally injected with $0.5\text{-}1 \times 10^4$ pfu MCMV one day after the NK cell transfer.

### Preparation of NK cells

Blood was collected via retro-orbital venous plexus by using heparinized capillary tubes (Thermo Fischer Scientific, Waltham, Massachusetts, U.S.A.). Splenocytes were incubated with biotinylated antibodies against CD4 (GK1.5), CD5 (53-7.3), CD8 (53-6.7), CD19 (6D5), Gr-1 (RB6-8C5), TER-119 (TER-119) (BioLegend San Diego, California, U.S.A.) and NK cells were enriched by negative selection using Dynabeads MyOne Streptavidin C1 (Thermo Fisher Scientific). Blood and splenic NK cells were defined as NK1.1[+]TCRβ[-]NKp46[+] lymphocytes unless stated otherwise. In some experiments, enriched NK cells from tamoxifen-administered NK-CreERT2 mice were stained with fluorochrome-conjugated mAbs against TCRβ (H57-597) and B220 (RA3-6B2) (BioLegend), and then YFP[+]TCRβ[-]B220[-] lymphocytes were purified as NK cells by sorting with FACSAria III (BD Biosciences).

For the RNA-seq and ChIP-seq, naïve Ly49H[+] and Ly49H[-] NK cells (YFP[+]NK1.1[+]TCRβ[-]Ly49H[+] and YFP[+]NK1.1[+]TCRβ[-]Ly49H[-] lymphocytes, respectively) and effector Ly49H[+] and Ly49H[-] NK cells (YFP[+]NK1.1[+]TCRβ[-]Ly49H[+]KLRG1[high] and YFP[+]NK1.1[+]TCRβ[-]Ly49H[-]KLRG1[+ to high] lymphocytes, respectively) were purified from the enriched splenic NK cells from naïve and MCMV-infected NK-CreERT2 mice on day 10 after tamoxifen administration. Long-lived memory Ly49H[+] NK cells and cytokine-activated Ly49H[-] NK cells (YFP[+]NK1.1[+]TCRβ[-]Ly49H[+]KLRG1[high]Ly6C[-]DNAM-1[- to low] and YFP[+]NK1.1[+]TCRβ[-]Ly49H[-]KLRG1[+ to high] lymphocytes, respectively) were purified from the enriched splenic NK cells from MCMV-infected NK-CreERT2 mice on day 28 after tamoxifen. These NK cells were purified by double sorting with FACSAria III (BD Biosciences).

### Co-transfer of Ly49H[+] NK cells and MCMV infection

*Tyrobp*[-/-] mice received $1 \times 10^5$ CD45.1[+] WT Ly49H[+] NK cells and either $1 \times 10^5$ CD45.2[+] *Themis2*[-/-] or *Zfp740*[-/-] Ly49H[+] NK cells and then infected with MCMV, as described previously[22].

To analyze the secondary expansion of memory NK cells, *Tyrobp*[-/-] mice received $1.5 \times 10^6$ CD45.1[+] WT Ly49H[+] NK cells and $1.0 \times 10^6$ CD45.2[+] *Themis2*[-/-] Ly49H[+] NK cells and then infected with MCMV. On day 28 pi, donor memory CD45.1[+] WT and CD45.2[+] *Themis2*[-/-] NK cells were enriched from the spleen of the recipient mice as described above. Naïve *Tyrobp*[-/-] mice received $3.0 \times 10^5$ memory CD45.1[+] WT and CD45.2[+] *Themis2*[-/-] NK cells and then infected with MCMV.

### Host protection assay against MCMV infection

*Tyrobp*[-/-] mice received $1 \times 10^5$ naïve Ly49H[+] NK cells purified from *Themis2*[+/+] or *Themis2*[-/-] NK-CreERT2 mice on day 10 after tamoxifen administration or $1 \times 10^5$ memory NK cells purified from MCMV-infected *Themis2*[+/+] or *Themis2*[-/-] NK-CreERT2 mice on day 28 after tamoxifen, and then infected with MCMV. MCMV titers were measured in the spleen and liver on day 3 after MCMV infection as described previously[23]. The copy number of MCMV IE1 gene was determined by quantitative PCR analysis with a 7500 Fast real-time PCR system and 7500 software v2.0.6 with Power SYBR Green PCR Master Mix (Thermo Fisher Scientific) and the following primers: forward, 5′-AGCCACCAAC ATTGACCACGCAC-3′; and reverse, 5′-GCCCCAACCAGGACACAC ATCTC-3′ (Hokkaido System Science, Sapporo, Hokkaido, Japan). A pDrive vector (Qiagen) encoding MCMV IE1 exon 3 was used for the standard curve.

### Flow cytometry

Cells were pretreated with Fc Block (2.4G2, BD Biosciences) before staining with the indicated fluorochrome-conjugated mAbs or isotype-matched mAbs (BD Biosciences, BioLegend, Thermo Fisher Scientific). mAbs used were FITC-conjugated anti-CD11b (M1/70), anti-CD27 (LG.3A10), anti-CD45.2 (104), anti-CD49b (DX5), anti-TCRβ (H57-597), PE-conjugated anti-CD11b (M1/70), anti-Ly49H (3D10), PerCPCy5.5-conjugated anti-NK1.1 (PK136), APC-conjugated anti-B220 (RA3-6B2), anti-CD45.2 (104), anti-CD49a (HMα1), anti-KLRG1 (2F1/KLRG1), anti-Ly6C (HK1.4), Alexa Fluor (AF) 700-CD45.1 (A20), anti-Ly-6C (HK1.4), anti-Ly6G (1A8), anti-NKp46 (29A1.4), PECy7-conjugated anti-B220 (RA3-6B2), anti-Ly6C (AL-21), anti-TCRβ (H57-597), Pacific Blue-conjugated anti-CD3ε (145-2C11), anti-CD69 (H1.2F3), Brilliant Violet (BV) 421-conjugated anti-KLRG1 (2F1/KLRG1), BV711-conjugated anti-CD11b (M1/70), biotinylated anti-DNAM-1 (TX42.1)[22], anti-Ly49H (3D10), anti-TCRβ (H57-597), and BV605-conjugated streptavidin.

For staining of intracellular Themis2, splenocytes from WT and *Themis2*[-/-] mice were stained with lineage marker-specific mAbs or naïve, effector, and long-lived NK cells from naïve and MCMV-infected *Themis2*[+/+] or *Themis2*[-/-] NK-CreERT2 mice on day 10 and 28 after tamoxifen administration were stained with fluorochrome-conjugated anti-Ly49H (3D10), anti-NK1.1 (PK136), anti-TCRβ (H57-597), and anti-KLRG1 (2F1/KLRG1) mAbs. Cells were fixed with 4% paraformaldehyde (Fujifilm Wako Pure Chemical Corporation, Tokyo, Japan) and permeabilized with 0.5% Triton X-100 (Fujifilm Wako Pure Chemical Corporation) at room temperature (RT), followed by blocking with 5% FCS in PBS at RT. Cells were washed with Intracellular Staining Perm Wash Buffer (BioLegend), incubated with anti-human/mouse/rat THEMIS2 rabbit IgG (ABclonal, Woburn, Massachusetts, U.S.A.) for 12 h at 4 °C, and stained with PE- or AF647-conjugated anti-rabbit IgG (Poly4064, BioLegend). The amount of Themis2 protein was represented as delta mean fluorescence intensity (MFI) calculated by subtracting MFI in *Themis2*[-/-] NK cells from that in WT or *Themis2*[+/+] NK cells (ΔMFI = MFI WT or *Themis2*[+/+] NK cells−MFI *Themis2*[-/-]).

For staining of intracellular active Caspase 3 and or 7, NK cells from the spleen of naïve or MCMV-infected *Tyrobp*[-/-] mice received $2.0 \times 10^5$ CD45.1[+] WT Ly49H[+] NK cells and $2.0 \times 10^5$ CD45.2[+] *Themis2*[-/-]

Ly49H⁺ NK cells were enriched on days 7, 17, and 28 pi, as described above, and then active Caspase 3 and or 7 were stained by using a FAM FLICA Caspase-3/7 Kit according to the manufacturer's instructions (Bio-Rad, Hercules, California, U.S.A.). Cells were washed and extracellular molecules were stained with fluorochrome-conjugated mAbs.

For staining of intracellular Ki-67, Bcl-2, and Bim, NK cells from the spleen of naïve or MCMV-infected *Tyrobp*⁻/⁻ mice received 2.0 ×10⁵ CD45.1⁺ WT Ly49H⁺ NK cells and 2.0 ×10⁵ CD45.2⁺ *Themis2*⁻/⁻ Ly49H⁺ NK cells were enriched on days 7, 17, and 28 pi, as described above, and then extracellular molecules were stained with fluorochrome-conjugated mAbs. Cells were fixed and permeabilized with eBioscience Foxp3 / Transcription Factor Staining Buffer Set (Thermo Fischer Scientific) for 30 min at RT. Cells were washed and incubated with AF647-conjugated anti-human/mouse Ki-67 (B56, BD Biosciences), PE-conjugated anti-mouse Bcl-2 (BCL/10C4, BioLegend), and AF488-conjugated anti-human/mouse/rat Bim (C34C5, Cell Signaling Technology, Danvers, Massachusetts, U.S.A.), or AF647-conjugated mouse IgG1 (MOPC-21, BioLegend), PE-conjugated mouse IgG1 (MOPC-21, BD Biosciences), and AF488-conjugated rabbit IgG (DA1E, Cell Signaling Technology) for 1 h at RT. The expression of Bcl-2 and Bim was represented as MFI.

For human NK cells, peripheral blood mononuclear cells were pretreated with FcR Blocking Reagent (Miltenyi Biotec, Bergisch Gladbach, North Rhine-Westphalia, Germany) and stained with FITC-conjugated anti-CD3 (HIT3α), anti-CD19 (HIB19), PE-conjugated anti-CD16 (3G8), and V450-conjugated anti-CD56 (B159) (BD Biosciences). The dilution factors for these antibodies for flow cytometry were 50 to 1000 or these antibodies were diluted and used according to the manufacturers' instructions.

In all experiments, doublet cells and dead cells were excluded by FSC-A-H gating, followed by SSC-A-W gating, and dead cells were excluded by using 510/50 or 525/50 nm filter of flow cytometers.

Samples were run on LSRFortessa and FACSAria III (BD Biosciences), data were obtained by using BD FACSDiva V8, and the data were analyzed with FlowJo V10 (FlowJo, Ashland, Oregon, U.S.A.).

## Ex vivo stimulation of NK cells

Naïve NK cells were enriched from the spleen of WT and *Themis2*⁻/⁻ mice, as described above, and primed with 1000 U/ml recombinant human IL-2 (BD Biosciences) in RPMI-1640 medium (Thermo Fisher Scientific) supplemented with 10% FCS (Thermo Fisher Scientific), 50 μM 2-mercaptoethenol, 2 mM L-glutamine, 100 U penicillin, 0.1 mg/mL streptomycin (Sigma-Aldrich), 10 mM HEPES, 1 mM sodium pyruvate, and 100 μM MEM non-essential amino acids (Thermo Fisher Scientific) (hereinafter, R10CM) for 16–24 h at 5% $CO_2$ and 37 °C. For stimulation of activating NK receptors, NK cells ($1 \times 10^5$) were stimulated with the following plate-bound mAbs (2.5 μg); mouse IgG1 (MOPC-21, BioLegend); mouse IgG2a (MOPC-173, BioLegend); rat IgG1 (R3-34, BD Biosciences); rat IgG2a (R35-95, BD Biosciences); anti-Ly49H mAb (3D10, BioLegend); anti-NKG2D (CX5, BioLegend); anti-NKp46 (29A1.4, BioLegend); and anti-NK1.1 (PK136, ATCC), as described previously[19]. Cells were cultured for 5 h at 5% $CO_2$ and 37 °C in R10CM in the presence of GolgiStop (BD Biosciences) and APC-conjugated anti-mouse CD107a mAb (1D4B, BioLegend), stained for surface molecules, fixed and permeabilized with Cytofix/Cytoperm (BD Biosciences), washed with Intracellular Staining Perm Wash Buffer (BioLegend), and stained with PE-conjugated anti-mouse IFN-γ mAb (XMG1.2, BioLegend). Degranulation of NK cells was determined by the frequency of CD107a⁺ cells by flow cytometry.

For in vitro assays for NK cell function, naïve NK cells were enriched from the spleen of naïve CD45.1⁺ WT, CD45.2⁺ *Themis2*⁻/⁻, and CD45.2⁺ *Zfp740*⁻/⁻ mice. For the preparation of memory NK cells, *Tyrobp*⁻/⁻ mice received 4.5-15 × 10⁵ CD45.1⁺ WT Ly49H⁺ NK cells and either 3.5-15 × 10⁵ CD45.2⁺ *Themis2*⁻/⁻ or *Zfp740*⁻/⁻ Ly49H⁺ NK cells, and then infected with MCMV. On day 28 pi, donor NK cells were enriched

from the spleen of infected recipient *Tyrobp*⁻/⁻ mice and used as memory NK cells. Naïve WT Ly49H⁺ NK cells were mixed with either naïve *Themis2*⁻/⁻ or *Zfp740*⁻/⁻ Ly49H⁺ NK cells. Similarly, memory WT Ly49H⁺ NK cells were mixed with either memory *Themis2*⁻/⁻ or *Zfp740*⁻/⁻ Ly49H⁺ NK cells. These mixed Ly49H⁺ NK cells were primed with 1000 U/ml recombinant human IL-2 (BD Biosciences) in R10CM. One to 5 × 10⁶ mixed naïve and memory Ly49H⁺ NK cells were co-cultured with 1-5 × 10⁵ RMA or RMA-m157 at a 1:1 ratio and degranulation and IFN-γ production of Ly49H⁺ NK cells was evaluated.

## Phosphorylation of ZAP70 and or Syk

NK cells were enriched from the spleen of WT and *Themis2*⁻/⁻ mice as described above, stained with fluorochrome-conjugated mAbs against CD49b (DX5), NKp46 (29A1.4), TCRβ (H57-597) (BD Biosciences), and resuspended with RPMI-1640 medium. NK cells were stimulated with 1 μg plate-bound anti-Ly49H mAb (3D10, BioLegend) at 37 °C for the indicated time, fixed with Cytofix/Cytoperm (BD Biosciences), permeabilized with Phosflow Perm Buffer III (BD Biosciences), followed by staining with PE-conjugated mAb against human or mouse phosphorylated ZAP70 (Y319)/Syk (Y352) (17 A/P-ZAP70, BD Biosciences). Phosphorylated ZAP70/Syk⁺ NK cells was evaluated by flow cytometry.

## Cloning and construction of Themis2 and Zfp740

For cloning an open reading frame (ORF) of human *THEMIS2*, NK cells in the blood of a healthy volunteer were enriched using CD56 MicroBeads (Miltenyi Biotec) and NK cells (CD3⁻CD19⁻CD56^dimCD16⁺ lymphocytes) were sorted. NK cells were dissolved in a TRIzol reagent (Thermo Fisher Scientific), RNA was isolated, and first-strand DNA was synthesized using a High-Capacity cDNA Reverse Transcription Kit (Thermo Fisher Scientific). The full-length *THEMIS2* ORF (UniProt Q5TEJ8-1) was amplified using a Platinum Taq (Thermo Fisher Scientific) and the following primers: forward, 5'-GACCATGGAGCCG GTGCCGCTGCAG-3'; and reverse, 5'-CCTCCAGCACTTAGATGGTT TTCTG-3' (Hokkaido System Science). The amplicon was ligated into pcDNA3.4 (Thermo Fisher Scientific) (pcDNA3.4-THEMIS2) and pEGFP-C1 (Takara Bio, Kusatsu, Shiga, Japan) (pEGFP-C1-THEMIS2).

3x FLAG tag was added at the C-terminus of the full-length *THEMIS2* ORF and constructed into pMXs-IRES-GFP retroviral vector (Cell Biolabs, San Diego, California, U.S.A) using a NEBuilder HiFi DNA Assembly Cloning Kit (New England BioLabs, Ipswich, Massachusetts, U.S.A.). A retroviral vector pMXs-IRES-GFP encoding FLAG-tagged THEMIS2 (THEMIS2-FLAG) was prepared by transfection of the retroviral construct and pcDNA3.4 encoding vesicular stomatitis virus G envelope glycoprotein were transfected into packaging 293gp cells (RIKEN BioResource Research Center, Tsukuba, Ibaraki, Japan) using Lipofectamine 3000 (Thermo Fischer Scientific). The retroviral supernatant was added into the culture of NKL-Ly49H cells in RetroNectin-coated wells (Takara Bio) and centrifuged at 700 x *g* rpm and 32 °C for 30 min. GFP⁺ NKL-Ly49H cells were sorted with BD FACS Aria III (BD Biosciences) and NKL-Ly49H cells expressing THEMIS2-FLAG was established. These procedures were approved by the Ethics Committee for Medical Sciences at the University of Tsukuba (approval number 234-2). All participants underwent an informed consent process.

For cloning an ORF of mouse *Themis2*, first-strand DNA was prepared from WT splenocytes. The full-length *Themis2* ORF (UniProt Q91YX0) was amplified by using a Platinum Taq (Thermo Fisher Scientific) and the following primers: forward, 5'-GACCATGGAGC CGGTGCGGCTGCAGGAC-3'; and reverse, 5'-TTCAAATTTCTTCAT AGTCATGGTCATC-3' (Hokkaido System Science). The amplicon was ligated into pcDNA3.4 (pcDNA3.4-Themis2). 3x FLAG tag was added at the C-terminus using a NEBuilder HiFi DNA Assembly Cloning Kit (New England BioLabs) (pcDNA3.4-Themis2-FLAG). Additional NLS of c-MYC was added at the N-terminus of Themis2-FLAG (pcDNA3.4-NLS-Themis2-FLAG).

For cloning an ORF of mouse *Zfp740*, first-strand DNA was prepared from splenic NK cells. The full-length *Zfp740* ORF (UniProt Q6NZQ6-1) was amplified using a Platinum Taq (Thermo Fisher Scientific) and the following primers: forward including MYC tag at the N-terminus, 5′-ATGGAACAAAAACTCATCTCAGAAGAGGATCTGATGCT GAGCCAGATTGC-3′; and reverse, 5′-CTATAGAGAAAACTGCCCT TCAG-3′ (Hokkaido System Science). The amplicon was ligated into pcDNA3.4 (pcDNA3.4-Zfp740-MYC).

## Subcellular localization of Themis2 in NK cells
A human NK cell line NKL-Ly49H was cultured in R10CM supplemented with 200 U/ml recombinant human IL-2 at 5% CO2 and 37 °C[55]. Cells were transfected with pEGFP-C1-THEMIS2 encoding GFP-fused THE-MIS2 by using a SF Cell Line 4D X Kit and 4D-Nucleofecton System (program DN-100) (Lonza, Basel, Switzerland). Cells were stimulated with plate-bound mouse IgG1 (MOPC-21) or anti-Ly49H mAb (3D10) (BioLegend) for 6 to 48 h, fixed with 4% paraformaldehyde, permeabilized with 0.5% Triton X-100, blocked with 1% bovine serum albumin (BSA) (Fujifilm Wako Pure Chemical Corporation), and then stained with anti-GFP rabbit mAb (D5.1, dilution factor 75, Cell Signaling Technology) for 1 h at RT, followed by AF488-conjugated anti-rabbit IgG (dilution factor 1000, Thermo Fisher Scientific).

Naïve Ly49H+ NK cells, effector Ly49H+ NK cells, and memory NK cells were purified from the spleen of naïve and MCMV-infected and tamoxifen-administered *Themis2*+/+ and *Themis2*−/− NK-CreERT2 mice, as described above. These NK cells were fixed, permeabilized, and blocked as described above. Cells were incubated with 85-340 ng rabbit anti-human/mouse THEMIS2 (Abcam, Cambridge, U.K.) for 36 h at 4 °C, followed by 10-50 ng AF647-conjugated anti-rabbit IgG (Poly4064, BioLegend).

Cells were cytospun by using a Shandon Cytospin 3 Centrifuge (Thermo Fisher Scientific), mounted with VECTASHIELD Vibrance Antifade Mounting Medium with DAPI (Vector Laboratories, Burlingame, California, U.S.A.), and observed under a confocal laser microscopy FluoView FV10i (Olympus, Tokyo, Japan). Percentages of nuclear Themis2 per cell were calculated by FLUOVIEW Ver4.1 (Olympus) and represented as a proportion of Themis2 signals overlapping DAPI relative to all Themis2 signals in a cell.

## RNA-seq
Naïve Ly49H+ and Ly49H- NK cells, effector Ly49H+ and Ly49H- NK cells, memory NK cells, and cytokine-activated Ly49H- NK cells were purified by double sorting with FACSAria III (BD Biosciences) as described above. RNA in sorted NK cells ($1.2 \times 10^3$ to $1.8 \times 10^5$) was isolated using a TRIzol reagent (Thermo Fisher Scientific), the yield was monitored using an Agilent Bioanalyzer RNA 6000 Pico Kit (Agilent Technologies, Santa Clara, California, U.S.A.), and processed to sequencing library using a SMART-seq Stranded Kit (Takara Bio). Sequencing was performed by NextSeq 500 (Illumina, Santa Monica, California, U.S.A.). FASTQ files were imported to CLC Genomics Workbench 12.0 (Qiagen, Hilden, Germany) and mapped to mm10. Normalized expression values were obtained by quantile normalization for the total count. DEGs were defined as having normalized expression >10 and $\log_2$ (fold change) >1 with false discovery rate (FDR) < 0.05. The pairwise heatmap of the number of DEGs and the heatmap of memory DEGs with relative expression with Z-score was drawn by using Heatmapper 1.0.0 (http://www.heatmapper.ca/).

## ChIP-seq
Naïve and memory NK cells ($1 \times 10^5$) were purified by sorting with FACSAria III (BD Biosciences) and fixed with 1% formaldehyde (Sigma-Aldrich). ChIP assays were performed using the following combination of antibodies and SDS concentration; anti-H3K4me3 Ab (ab8580, Abcam, 0.1% SDS); and anti-H3K27ac Ab (ab4729, Abcam, 0.2% SDS), as

described previously[56]. ChIP DNA was processed into a DNA sequencing library using NEBNext Ultra II DNA Library Prep Kit for Illumina (New England Biolabs) and paired-end sequencing was performed with NextSeq 500 (Illumina). H3K4me3 and H3K27ac reads of naïve and memory WT and *Themis2*−/− NK cells were subtracted with input controls and unique peaks were identified (16,466-20,510 H3K27ac ChIP-seq peaks and 14,109-18,345 H3K4me3 ChIP-seq peaks in naïve Ly49H+ NK cells and 14,756-20,764 H3K27ac ChIP-seq peaks and 13,301-17,703 H3K4me3 ChIP-seq peaks in memory NK cells) using MACS3 3.0.0a6 (https://github.com/macs3-project/MACS).

H3K4me3 DARs uniquely present in either memory *Themis2*+/+ or *Themis2*−/− NK cells were extracted by bedtools on Galaxy v21.09[57] and these unique H3K4me3 DAR-associated single nearest genes (memory H3K4me3 DAR genes) were identified by using GREAT v4.0.4[58]. H3K27ac DARs uniquely present in either memory *Themis2*+/+ or *Themis2*−/− NK cells were extracted by Galaxy. To highlight non-promoter H3K27ac-marked regions, these unique H3K27ac DARs were overlapped with CTCF-bound enhancer-like signatures (ELS) available in ENCODE (https://www.encodeproject.org/)[59], because CTCF binding correlates with enhancer activity and enhancer-promoter interaction to maintain gene expression[60]. These unique H3K27ac DAR-associated single nearest genes with distal and proximal ELS (memory H3K27ac DAR distal and proximal ELS genes, respectively) were identified by using GREAT. The heatmap of memory H3K4me3 DARs with the relative read count was drawn by Morpheus (https://software.broadinstitute.org/morpheus/). H3K4me3 and K3K27ac reads of naïve and memory NK cells versus input controls were normalized and scaled by Galaxy. These epigenome-wide heatmaps of the read density in 5 kb from transcription start sites (TSS) were drawn by deepTools on Galaxy. The heatmaps of pairwise intersections of individual and unified atlases of H3K4me3 peaks and H3K27ac peaks of naïve and memory NK cells was drawn with their Spearman's correlations for pairwise Jaccard statistics by deepTools on Galaxy. The distance from individual memory DARs to TSS was plotted with the number of DARs within each distance range by GREAT.

## Bioinformatics
GO analysis of memory DEGs was performed by DAVID v2021q4[61] and ShinyGO v0.6.1[62]. GO analysis of memory DAR genes were performed by GREAT.

Functional protein association networks of mouse Themis2 and human THEMIS2 were constructed and KEGG Pathways of their networks were implemented by STRING[63].

Principal component analysis (PCA) of naïve *Themis2*+/+ and *Themis2*−/− Ly49H+ NK cells and memory *Themis2*+/+ and *Themis2*−/− NK cells was performed by using iDEP .96[64].

GSEA V4.2.3[65] was performed by using normalized expression in memory WT and *Themis2*−/− NK cells and normalized enrichment score (NES) and FDR were computed.

For an integrative analysis of memory DEGs and memory H3K4me3 DARs, the absolute values of the $\log_2$-transformed fold change of normalized expression of memory DEGs and the fold change of signal per million reads of memory H3K4me3 DARs were represented as scatter plots. Spearman's correlation coefficients with their *p* values of all memory DEGs/ memory H3K4me3 DARs and these genes annotated with the GO term DNA-templated transcription were calculated by using cor.test() function in R (https://www.r-project.org/). A mouse gene list annotated with DNA-templated transcription was obtained by using biomaRt package in R.

The Venn diagram was drawn by Venny 2.0 (https://bioinfogp.cnb.csic.es/tools/venny/).

The knowledge-based network analysis on ZNF740 was performed by Ingenuity Pathway Analysis 20.0[66] and the protein-protein interactions in the nucleus were represented.

## Motif analysis

Transcription factor binding motifs in promoters of memory DEGs and memory H3K4me DAR genes were analyzed by the MEME Suite 5.3.0[67]. Enriched motifs in promoter sequences (200 bp upstream and 100 bp downstream of TSS) were discovered by MEME STREME. Transcription factors that can bind to these motifs were ranked by MEME Tomtom. The $p$ value for each motif was calculated as the probability that a random motif of the same width as the target would have an optimal alignment with a match score as good or better than the target's. MEME Tomtom estimates the $p$ value using a null model consisting of sampling motif columns from all the columns in the set of target motifs[68].

## Identification of Themis2-binging nuclear proteins by mass spectrometry

To identify Themis2-binding nuclear proteins by pull-down with for expression of FLAG-tagged Themis2 protein (Themis2-FLAG), we used effector $Themis2^{-/-}$ NK cells as a source of nuclear proteins. Effector $Themis2^{-/-}$ NK cells were purified from the spleen of MCMV-infected and tamoxifen-administered $Themis2^{-/-}$ NK-CreERT2 mice on day 10 pi. Sorted NK cells were homogenized in hypotonic buffer (10 mM HEPES, 10 mM NaCl, 1.5 mM $MgCl_2$ (Sigma-Aldrich), protease inhibitor cocktail (Merck Millipore, Burlington, Massachusetts, U.S.A.)) by using a Dounce homogenizer (Wheaton Industries, Millville, New Jersey, U.S.A.). The nuclear pellets were lysed in nuclear lysis buffer (0.1% Nonidet P-40 (nacalai tesque, Kyoto, Japan), 50 mM HEPES, 150 mM NaCl (Sigma-Aldrich), protease inhibitor cocktail (Merck Millipore)) by sonication and the supernatants were collected as nuclear proteins.

To prepare Themis2-FLAG, pcDNA3.4-Themis2-FLAG was transfected into 293 F cells using an ExpiFectamine 293 Transfection Kit (Thermo Fisher Scientific). On day 3 post-transfection, 293 F cells were homogenized in hypotonic buffer. Lysates were incubated with anti-FLAG M2 affinity gel (Sigma) in HEPES-buffered saline supplemented with protease inhibitor cocktail (Merck Millipore) for 2 h at 4 °C with agitation and eluted with 3x FLAG peptides (Sigma-Aldrich) by incubating 30 min at 4 °C. The purity and quality (e.g. monomer, aggregation) was confirmed by SDS-PAGE, followed by silver staining using a Pierce Silver Stain Kit (Thermo Fisher Scientific).

Two-milligram nuclear proteins of effector $Themis2^{-/-}$ NK cells were incubated with 200 μg Themis2-FLAG for 3 h at 4 °C with agitation. The mixtures were chemically crosslinked with 1 mM 3,3'-dithiobis(sulfosuccinimidyl propionate, DTSSP) (Thermo Fisher Scientific) for 30 min on ice. After gel filtration of the mixtures by using a HiTrap Desalting column (Cytiva, Marlborough, Massachusetts, U.S.A.), Themis2-FLAG-bound proteins were immunoprecipitated with anti-FLAG M2 affinity gel (Sigma-Aldrich) and eluted with 3x FLAG peptides (Sigma-Aldrich). The immunoprecipitants were separated by SDS-PAGE by using Mini-PROTEAN TGX polyacrylamide (PA) gels (Bio-Rad), followed by silver staining using a Pierce Silver Stain for Mass Spectrometry (Thermo Fisher Scientific). Selected protein bands that appeared only in immunoprecipitants with Themis2-FLAG and blank gels with the same molecular weight positions were excised and proceeded previously as described[69]. The resulting peptide peaks uniquely present in selected gel pieces, but not in blank gel pieces, were picked up and proteins were identified through peptide mass fingerprint matching by using flexControl 3.4 and MASCOT Server 2.7 with UniProtKB/Swiss-Prot database (https://www.uniprot.org/). Transcription factors and epigenetic enzymes among Themis2-bound nuclear proteins were filtered; those expressed in NK cells were extracted by utilizing RNA-seq data of NK cell subsets and were represented.

## Biochemistry

NKL-Ly49H cells expressing THEMIS2-FLAG were stimulated with anti-Ly49 mAb for 2 or 5 min, as described above. Stimulated cells were homogenized in hypotonic buffer and the supernatants were collected as cytosolic proteins, as described above. Cytosolic proteins were treated with or without DTSSP and THEMIS2-FLAG-bound proteins were immunoprecipitated with anti-FLAG M2 affinity gel (Sigma-Aldrich) and eluted with 3x FLAG peptides (Sigma-Aldrich). The immunoprecipitants were separated by SDS-PAGE, transferred into polyvinylidene difluoride membranes (Merck Millipore), and immunoblotted with rabbit anti-DAP12 mAb (D7G1X, Cell Signaling Technology) or rabbit anti-ZAP70 mAb (99F2, Cell Signaling Technology) for 12–16 h at 4 °C, followed by HRP-conjugated anti-rabbit IgG (BioLegend). The chemiluminescence was developed with a SuperSignal West Pico PLUS Chemiluminescence Substrate (Thermo Fisher Scientific) and detected with ImageQuant LAS 4000 mini and ImageQuant TL software ver8 (Cytiva). The membrane was reblotted with biotinylated anti-FLAG mAb (M2, Sigma-Aldrich), followed by streptavidin-HRP (Cytiva).

pmaxGFP (Lonza) alone as a mock, pcDNA3.4-NLS-Themis2-FLAG plus pmaxGFP, pcDNA3.4-Themis2-FLAG plus pcDNA3.4-Zfp740-MYC, and pcDNA3.4-NLS-Themis2-FLAG plus pcDNA3.4-Zfp740-MYC were co-transfected into 293 T cells using a ProFection Mammalian Transfection System (Promega). On day 2 post-transfection, 293 T cells were homogenized in hypotonic buffer and the supernatants were collected as cytosolic proteins. The nuclear pellets were lysed in nuclear lysis buffer by sonication and the supernatants were collected as nuclear proteins.

Themis2-FLAG-binding nuclear proteins were immunoprecipitated by incubation with anti-FLAG M2 affinity gel (Sigma-Aldrich) and eluted with 3x FLAG peptides (Sigma-Aldrich). The immunoprecipitants were separated by SDS-PAGE, transferred into polyvinylidene difluoride membranes (Merck Millipore), and immunoblotted with mouse anti-MYC-tag mAb (9B11, Cell Signaling Technology) for 16 h at 4 °C, followed by HRP-conjugated anti-mouse IgG (Cytiva). The chemiluminescence was developed with a SuperSignal West Pico PLUS Chemiluminescence Substrate (Thermo Fisher Scientific) and detected with ImageQuant LAS 4000 mini (Cytiva). The membrane was reblotted with biotinylated anti-FLAG mAb (M2, Sigma-Aldrich), followed by streptavidin-HRP (Cytiva). The nuclear translocation efficiency of Themis2-FLAG and NLS-Themis2-FLAG was compared by calculating the ratio of nuclear to cytosolic FLAG signals.

The dilution factors for these antibodies for biochemistry were 1000 to 2000 or these antibodies were diluted and used according to the manufacturers' instructions.

## Luciferase reporter assay

PGK promoters harboring Zfp740-binding motif (CCCCCCCAC) or random sequences were amplified by using an Ex Taq (Takara Bio), pRetroQ-DsRed Monomer-N1 (Takara Bio) as a template of PGK promoter, and the following primer pairs including Zfp740-binding motif or random sequences: Zfp740 motif forward, 5'-ATGCCTCGAGCCCCCCCACGGGTAGGGGAGGCGCTTTTC-3'; Zfp740 motif reverse, 5'-ATGCAAGCTTGTGGGGGGGCGAAAGGCCCGGAGATGAGG-3'; random forward, 5'-ATGCCTCGAGATGCATGCAGGGTAGGGGAGGCGCTTTTC-3'; and random reverse, 5'-ATGCAAGCTTATGCATGCACGAAAGGCCCGGAGATGAGG-3' (Hokkaido System Science). Amplified promoters were ligated into a luciferase reporter vector PGV-B2 (TOYO B-net, Tokyo, Japan) (hereinafter, PGV-Zfp740 motif and PGV-random).

To evaluate the transcriptional activity of Zfp740, 293 T cells were cultured in 24-well culture plates, and either pcDNA3.4-Zfp740 or PGV-B2 (25 ng) and either PGV-Zfp740 motif or PGV-random (100 ng) were co-transfected with pmaxGFP (100 ng, Lonza) using a ProFection Mammalian Transfection System (Promega). To evaluate an effect of nuclear Themis2 on the transcriptional activity of Zfp740, either pcDNA3.4-Zfp740 or PGV-B2 (5 ng) and either pcDNA3.4-NLS-Themis2 or PGV-B2 (25 ng) and either PGV-Zfp740 motif or PGV-random (100 ng) were co-transfected with pmaxGFP. At 36 h post-

transfection, luciferase-mediated luminescence was developed with a BrilliantStar-LT reagent (TOYO B-net) and the luminescence and GFP fluorescence were measured with Varioskan LUX (Thermo Fisher Scientific) and Skanlt RE software 6.1.1. The transcriptional activity was represented as relative light unit of the luminescence normalized by the fluorescence.

## Quantitative reverse transcription (qRT)-PCR

Naïve, effector, and long-lived NK cell subsets were sorted, first-strand DNA was synthesized, and qRT-PCR was performed on an ABI 7500 Fast real-time PCR system with ABI Power SYBR Green PCR Master Mix (Thermo Fisher Scientific) and the following primers: *Gapdh* forward, 5′-TGGTGAAGGTCGGTGTGAAC-3′; *Gapdh* reverse, 5′-ATGAAGGGG TCGTTGATGGC-3′; *Bcl10* forward, 5′-CTTCAAGTAGAAAACGGGCTGG-3′; *Bcl10* reverse, 5′-GCACCTAGAGAGGTTGTTGGT-3′; *Map4k4* forward, 5′-CTGGCCGCCATCAAGGTTAT-3′; *Map4k4* reverse, 5′-AGCAC-CATAGTACGTGGCAAT-3′; *Niban2* forward, 5′-ATGGGAGACGTA CTGTCCACA-3′; *Niban2* reverse, 5′-TCCTCATAGAAGCGAAGGAACT-3′; *Sirt1* forward, 5′-TGATTGGCACCGATCCTCG-3′; *Sirt1* reverse, 5′-CCA-CAGCGTCATATCATCCAG-3′; *Themis2* forward, 5′-AAACTCTGCAC-CAAATCCTGC-3′; *Themis2* reverse, 5′- GCTTGGAGCATATACTG GGGTT-3′; *Zfp740* forward, 5′-ACCTCAAGAGGCACGTCCTAA-3′; and *Zfp740* reverse, 5′-CTTGTGGCGTTCGAGATGGT-3′ (Hokkaido System Science). Relative quantities of transcripts were normalized to those of *Gapdh* transcripts.

## Re-analysis of public database

Expression of *Zfp740* in the organs and immune cell types in mice was investigated by re-analysis of publicly deposited data (GSE10246 and GSE116177, respectively)[70,71].

## Statistical analysis

The two-sided Student's $t$ test and one-way analysis of variance (ANOVA) were used to compare the data by using GraphPad Prism (GraphPad Software, San Diego, California, U.S.A.). $p < 0.05$ was considered statistically significant. Error bars show standard deviations (SD).

## Reporting summary

Further information on research design is available in the Nature Portfolio Reporting Summary linked to this article.

## Data availability

The RNA-seq data, ChIP-seq data, and all the datasets generated in this study have been deposited in the GEO under the accession code GSE227139. Source data are provided with this paper.

## Code availability

All codes used for data analysis are available at https://github.com/CAB314/Themis2-Nabekura [https://doi.org/10.5281/zenodo.8373826].

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

## Acknowledgements

We are grateful to Mitsue Arimoto (University of Tsukuba, Japan) for conducting the experimental procedures and assisting in the MALDI-TOF mass spectrometry and MASCOT search. We are grateful to Lewis L. Lanier (University of California, San Francisco) for critical comments and by providing NK-CreERT2 mice. We are grateful to Richard J. Cornall (University of Oxford) by providing *Themis2*[-/-] mice. We are also grateful to Rei Hirochika and Seiya Mizuno (University of Tsukuba, Japan) for the genotyping of *Themis2*[-/-] mice and the generation of *Zfp740*[-/-] mice, respectively. We thank Eri Takenaka (University of Tsukuba) for preparing the salivary grand virus of MCMV. This work was supported by JSPS KAKENHI 16H06387, 18H05022, and 21H04836 (A.S.), 17H05071, JST FOREST JPMJFR2166, the Japanese Society for Immunology Outstanding Young Immunology Researcher Award, University of Tsukuba Seeding Program for Research, and Tsukuba Basic Research Support Program (T.N.).

## Author contributions

T.N. contributed to project planning, experimental work, data analysis, and writing of the manuscript. E.A.D. contributed to the experimental work. S.T. and Y.A. contributed to the data analysis. K.K., A.F., M.M., and P.E.L. contributed by providing essential materials and critical comments. A.S. contributed to the project planning and writing of the manuscript.

## Competing interests

The authors declare no competing interests.
