## [Peer Review File · Nature Communications]

Themis2 regulates natural killer cell memory function and formationREVIEWER COMMENTS

Reviewer #1 (Remarks to the Author):

The manuscript represents a significant amount of work, in which the authors have carefully applied standard techniques in developmental immunology to identify Themis2 and Zfp740 as novel negative regulators of NK memory in mice. Many of these experiments have been done on small numbers of primary cells, so will have posed significant technical challenges. The authors are to be congratulated on the work, and I support its publication.

One point that I think should be further explored is the extent to which Themis2 specifically regulates NK memory, as opposed to non-memory effector functions. In particular, I note that Figure 2b shows increased degranulation in naïve Themis2 KO NK cells, suggesting a role in effector functions more broadly. Interestingly, even though the Zfp740 KO phenocopies the Themis2 KO for most measurements, for degranulation there is no difference between WT and Zfp740 KO cells (Fig 6e). Does this suggest that the requirement for Zfp740 is more specific to memory than that of Themis2? If the authors have any further functional data from naïve NK cells from each of the KO, it might be useful to present this, but otherwise this should at least be discussed.

Minor points

Line 180. "...memory WT and Themis2^{-/-} NK cells were co-transferred into Tyrobp^{-/-} mice and infected with MCMV." Clarify that it was the mice (not the cells, as currently written) that were infected.

Figure 3. Data from the human NK-like cell line NKL is presented in panels A and B, presumably because of the relative tractability of this line. However, given that the more relevant experiment, using primary mouse cells, is also presented below, it's not clear what this adds: this should be made more explicit. In both the NKL and the primary mouse cell experiments, it appears that Themis2 expression as a whole increases following Ly49H stimulation (regardless of localisation). Can total expression be quantified to determine whether this is the case? Presumably, this is in line with the fact that Themis2 was initially identified as potentially important for NK memory formation because it is overexpressed in memory NK cells.

Reviewer #2 (Remarks to the Author):

Nabekura et. al. aim to shed light on the mechanism by which NK cell memory is developed. They do so by identifying a new negative regulator of NK cells, Themis2, that seems to be expressed in many cell stages and affects both the proliferation, the function and the differentiation of these cells. The authors recognize two main mechanisms by which Themis2 performs these roles; inhibiting downstream signaling from Ly49H, and assisting transcriptional repression of key genes by Zfp740.

The data presented by Nabekura et. al. is robust and mostly well-controlled, with some details and controls missing, as described in the following section.

The mechanism behind the formation of innate memory in general and NK cell memory specifically is a fascinating question and a central one in current immunology. The data presented here add details about a major regulator of this process and possibly of NK function in general. As such, this manuscript could be a significant addition to the field following some revisions.

Issues in detail

1. What is already known regarding Themis2 in the literature? It might benefit the reader if the authors would add some information regarding this into the introduction.
2. The mouse model the authors used that is described in Figure 1 is difficult to understand. The authors wrote that "we used NKp46-CreERT2 Tg mice with Rosa26-YFP/YFP alleles (hereinafter,

NK-CreERT2 mice), which allowed us to track MCMV-primed NK cells as YFP+ NK cells after tamoxifen administration following MCMV infection". These mice will have Cre expression in NKp46+ NK cells, and a constitutive expression of YFP. How can this identify MCMV-primed NK cells? Is NCR1 only expressed on NK cells after activation in mice? Why are only the Ly49H+ cells marked with YFP? The authors should provide a thorough explanation of their mouse model

3. Line 115-116: "Among three genes, we chose Themis2 as the focus of the present study because we confirmed expression in naïve NK cells at the protein level". In Fig. 1b the authors say that they chose candidate genes according to them being upregulated in activated or memory NK cells relative to naïve NK cells. Is Themis2 expressed in all NK cell subsets and is only quantitatively upregulated upon NK activation? Should the authors not validate its expression at the protein level on the relevant cells – activated and memory NK cells? And if Themis2 is indeed expressed to some extent in naïve NK cells, I think it is worth discussing its possible roles in them as part of the discussion.

4. From Fig. 1 it seems like Themis2 negatively regulates NK proliferation in all stages of activation and infection. Can the authors differentiate between its initial effects and its effects in later, memory-related states?

Is it possible that Themis2 KO allows for a larger pool of NK cells at the early stages of infection, and as a result more memory NK cells can emerge?

5. If Themis2 negatively regulates NK proliferation, do Themis2 KO mice have more naïve NK cells at baseline?

6. Extended data Fig. 3 - The authors present very convincing data showing that Themis2 negatively regulates Ly49H signaling via some mechanism involving either Zap70 or Syk phosphorylation. As this seems like a major way by which Themis2 can regulate NK cell function, it will be beneficial to describe it a bit further and recognize the specific target affected by Themis2. Can this target be DAP12 and not directly Zap70 or Syk?

Among other methods, this could be performed by either repeating the experiments shown in the abovementioned figure using specific antibodies for the different proteins or by co-immunoprecipitation experiments looking for direct interaction between Themis2 (or its Flag-tagged construct presented in fig.5), Zap70, Syk or the upstream DAP12 protein.

7. Extended data fig.3 – If Themis2 negatively regulates Zap70/Syk /DAP12 signaling, can it also negatively regulate other NK receptors using a similar signaling cascade (for example NKG2D, among others)? Can the authors repeat the experiments shown in the above-mentioned figure using activating antibodies against other such receptors? In any case, a control using an antibody that activates a receptor with a different DAP12-ZAP70-Syk-independent signaling cascade is needed to further validate these results.

8. Figure 5a – Only data from activated NK cells is shown, while Themis2 appears to be translated in memory NK cells and to some extent also in naïve NK cells (Extended data fig. 2a). The differential binding and precipitation of Themis2 to proteins in these two cell stages relative to activated NK cells and to each other is also important to the understanding of Themis2's role in NK cells and memory creation.

Technical issues

1. Lines 69-71: unclear sentence: "They recognize unhealthy cells by utilizing a repertoire of activating NK receptors and exert the effector function, i.e., cytotoxicity, interferon (IFN)- γ production, for the eradication of these cells".

Should it be their effector functions instead? Also, it can be understood from this sentence that NK cells only secrete IFN γ , which is of course only a single example of the several cytokines and chemokines they secrete as part of their effector functions.

2. Line 79: "Ly49H+ NK cells expand as effector NK cells, undergo apoptosis after the peak of NK cell response, and differentiate into long-lived memory NK cells during MCMV infection".

The sentence might be clearer if it will be "...undergo apoptosis after the peak of NK cell response, or differentiate into long-lived memory NK cells"

3. Extended data figure 2a,4a – The authors show a representation of 1 experiment. Can they also show the quantification of all repeats?

4. Fig. 3c+d: The cells described in figure 3a+b are human. Are the cells depicted in Fig. 3c+d human or mouse? How were they generated? These results would be clearer if this information will be presented in the text and/or in the figure legend.

Reviewer #3 (Remarks to the Author):

In their manuscript, Shibuya and colleagues provide evidence that Themis2 acts as a negative regulator of NK cell responses against MCMV infection. Using mass spec, they mechanistically determine a novel themis2 binding partner Zfp740, generate luciferase reporter lines to show Zfp740 is a repressor, and then make new Zfp740 ko mice to recapitulate the Themis2 phenotype. These impressive findings will be of broad interest to those in the field of immunology. I feel that with the edits I've suggested below, this manuscript would be a great fit for publication in Nature Communications.

Specific comments:

1. Is NK cell effector function (e.g., IFN γ production/degranulation) similarly enhanced in response to stimulation by other activating receptors (e.g., anti-NK1.1, anti-NKG2D, and/or anti-Ly49D)? Given that the hypothesis that Themis2 interaction may only be specific for activating receptors that utilize DAP12 as co-receptors, these data would add nuance the overall findings.

2. Themis2 is highly upregulated in effector and memory Ly49H⁺ NK cells compared to naïve cells. Is the enhanced effector function in these states associated with enhanced pZAP70/Syk as well? Do the authors see differences in IFN γ production on day 2 PI in vivo? A discussion on how the Calcium-NFAT signaling would be affected and contribute to the overall enhanced response in Themis2-deficient NK cells would be of interest.

3. While the authors implied enhanced survival in Themis2-deficient Ly49H⁺ NK cells, they have not demonstrated that the expansion and contraction advantage observed in these cells are due to cell survival versus proliferation. Assessing Ki-67, Bcl2, Bim, and/or Caspase3/7 at different time points during infection could provide further proofs that this phenotype is due to enhanced survival. CTV can also be used to assess differences in proliferation.

Minor:

1. How many genes showed differential expression between WT and Themis2-deficient Ly49H⁺ NK cells in each time point and pairwise comparison? Can PCA analysis be shown and included in the supplementary figure?

2. How many peaks were identified from H3K4me3 and H3K27Ac ChIP-seq between WT and Themis2-deficient NK cells at naïve and memory time points?

3. What do the authors mean by "pairwise intersection analyses"? Does it refer to comparing overlapping histone peaks between WT and Themis2-deficient NK cells? Was a unified atlas of all conditions used to generate this Enrichment heatmap plot? Or does it utilize individual atlas? If the latter, then please use the unified atlas instead of individual atlas. This might help reveal the differences in H3K4me3 and H3K27Ac signals.

4. Are the differentially expressed genes (DEGs) between WT and Themis2-deficient memory NK cells the ones that exhibit differences in H3K4me3 signals between WT and Themis2-deficient NK cells? Can the authors provide some examples and global integrative analysis of H3K4me3 and RNA-seq?

5. What is the rationale for overlapping CTCF binding sites with H3K27Ac? How was the CTCF binding atlas generated? Is it based on scanning the mouse genome with CTCF motifs? Or is it from NK cell CTCF ChIP-seq? If so, from what time point?

6. How many proteins are pulled down by this assay? And how many of those are not transcription factor? Given that Themis1/2 contain Proline-rich sequence (PRS) that binds to SH3 domain-containing proteins, such as Grb2, do the authors observed any of SH3 domain-containing proteins getting pulled down?

Responses to the Reviewer #1

The manuscript represents a significant amount of work, in which the authors have carefully applied standard techniques in developmental immunology to identify Themis2 and Zfp740 as novel negative regulators of NK memory in mice. Many of these experiments have been done on small numbers of primary cells, so will have posed significant technical challenges. The authors are to be congratulated on the work, and I support its publication.

We appreciate the reviewer's generous comment.

One point that I think should be further explored is the extent to which Themis2 specifically regulates NK memory, as opposed to non-memory effector functions. In particular, I note that Figure 2b shows increased degranulation in naïve Themis2 KO NK cells, suggesting a role in effector functions more broadly.

We agree with the reviewer that Themis2 inhibits activation signaling via Ly49H even in naïve NK cells (Figure 2a). In this revised manuscript, we showed that Themis2 suppressed NK cell activation after stimulation of NKG2D, Nkp46, and NK1.1 (NKR-P1C) (new Supplementary Figure 3c). Thus, Themis2 inhibits activation signaling through a diverse group of activating NK receptors in naïve NK cells.

We described these results in the sections of Results, Discussion, Materials, Methods, and Figure Legends in the revised manuscript (lines 141-143, 415-419, 880-883 in the main text, and lines 77-79 in Supplementary Figures).

Interestingly, even though the Zfp740 KO phenocopies the Themis2 KO for most measurements, for degranulation there is no difference between WT and Zfp740 KO cells (Fig 6e). Does this suggest that the requirement for Zfp740 is more specific to memory than that of Themis2? If the authors have any further functional data from naïve NK cells from each of the KO, it might be useful to present this, but otherwise this should at least be discussed.

As the reviewer pointed out, Zfp740 did not inhibit effector functions of naïve Ly49H⁺ NK cells, suggesting that Zfp740 deficiency has little impact on the function of naïve NK cells and might

be more specific to memory NK cells than Themis2. However, we do not have further functional data from naïve *Zfp740*^{-/-} NK cells.

We discussed this issue in the Discussion section of the revised manuscript (lines 423-431 in the main text).

Minor points

Line 180. "...memory WT and Themis2^{-/-} NK cells were co-transferred into Tyrobp^{-/-} mice and infected with MCMV." Clarify that it was the mice (not the cells, as currently written) that were infected.

We corrected the sentence to clarify that these mice were infected with MCMV in the revised manuscript (lines 213-214 in the main text).

Figure 3. Data from the human NK-like cell line NKL is presented in panels A and B, presumably because of the relative tractability of this line. However, given that the more relevant experiment, using primary mouse cells, is also presented below, it's not clear what this adds: this should be made more explicit.

We showed the *ex vivo* results of subcellular localization of Themis2 in naïve, effector, and memory Ly49H⁺ NK cells after MCMV infection in Figures 3c and 3d, suggesting that Ly49H stimulation induces the nuclear translocation of Themis2 in Ly49H⁺ NK cells. However, we still cannot exclude the possibility that inflammation following MCMV infection triggers the translocation via some undetermined signals other than Ly49H. To address this issue, we utilized NKL-Ly49H expressing GFP-fused Themis2 and stimulated by plate-bound anti-Ly49H *in vitro*, because primary NK cells were difficult to be used due to a technical limitation; a reliable and validated two different clones of anti-Ly49H mAbs are not available for both purification of Ly49H⁺ NK cells and stimulation of Ly49H on primary NK cells.

We explained the issue in the Results section (lines 226-228 and 238-239 in the main text).

In both the NKL and the primary mouse cell experiments, it appears that Themis2 expression as

a whole increases following Ly49H stimulation (regardless of localization). Can total expression be quantified to determine whether this is the case? Presumably, this is in line with the fact that Themis2 was initially identified as potentially important for NK memory formation because it is overexpressed in memory NK cells.

According to the reviewer's suggestion, we quantified total amounts of Themis2 protein in naïve, effector, and long-lived Ly49H⁺ and Ly49H⁻ NK cells *ex vivo* by intracellular staining of cytosolic and nuclear Themis2 protein by flow cytometry. In line with the kinetics of *Themis2* mRNA during MCMV infection (Supplementary Figures 2b and 2c), the amount of Themis2 protein was increased in effector Ly49H⁺ NK cells and maintained in memory NK cells during MCMV infection (new Supplementary Figure 2d).

We described these results in the sections of Results, Materials, and Methods, and Figure Legends in the revised manuscript (lines 126-128, 831-834, 840-843 in the main text, and lines 55-64 in Supplementary Figures).

Responses to the Reviewer #2

Nabekura et. al. aim to shed light on the mechanism by which NK cell memory is developed. They do so by identifying a new negative regulator of NK cells, Themis2, that seems to be expressed in many cell stages and affects both the proliferation, the function and the differentiation of these cells. The authors recognize two main mechanisms by which Themis2 performs these roles; inhibiting downstream signaling from Ly49H, and assisting transcriptional repression of key genes by Zfp740. The data presented by Nabekura et. al. is robust and mostly well-controlled, with some details and controls missing, as described in the following section.

The mechanism behind the formation of innate memory in general and NK cell memory specifically is a fascinating question and a central one in current immunology. The data presented here add details about a major regulator of this process and possibly of NK function in general. As such, this manuscript could be a significant addition to the field following some revisions.

We appreciate the generous comments by the reviewer.

Issues in detail

1. What is already known regarding Themis2 in the literature? It might benefit the reader if the authors would add some information regarding this into the introduction.

We appreciate the reviewer's suggestion. We added some information regarding Themis2 in the literature in the Results section, rather than the Introduction section (lines 120-122 in the main text).

2. The mouse model the authors used that is described in Figure 1 is difficult to understand. The authors wrote that "we used NKp46-CreERT2 Tg mice with Rosa26-YFP/YFP alleles (hereinafter, NK-CreERT2 mice), which allowed us to track MCMV-primed NK cells as YFP+ NK cells after tamoxifen administration following MCMV infection".

These mice will have Cre expression in NKp46+ NK cells, and a constitutive expression of YFP. How can this identify MCMV-primed NK cells? Is NCR1 only expressed on NK cells after activation in mice? Why are only the Ly49H+ cells marked with YFP? The authors should provide a thorough explanation of their mouse model.

We apologize for the inadequate information. We developed a transgenic mouse carrying the CreERT2 recombinase expressed under the control of the *Ncr1* gene (*J Exp Med* 2016;213:2745.). The NKp46-CreERT2 Tg mice were intercrossed with Rosa26-YFP B6 mice harboring a loxP-flanked STOP sequence followed by the YFP sequence into the *Gt(ROSA)26Sor* loci. Consequently, NK cells (precisely, NKp46+ cells) in NKp46-CreERT2 Tg mice with Rosa26-YFP/YFP alleles (hereinafter, NK-CreERT2 mice) express YFP upon CreERT2-mediated excision of the loxP-flanked STOP sequence in the Rosa26 loci only after tamoxifen administration. When these NK-CreERT2 mice are infected with MCMV and then treated with tamoxifen on days 0, 1, and 2 post-infection, we can label MCMV-primed NK cells, irrespective of NK cell subsets expressing or not expressing Ly49H, with YFP and track their fate *in vivo*. Therefore, we identified MCMV-primed (that is, YFP-positive) effector and long-lived

Ly49H⁺ and Ly49H⁻ NK cells with stringent phenotypic definition in accordance with our previous studies (*Immunity* 2014;20:225. and *J Exp Med* 2016;213:2745.).

The fate-tracking mouse model can label not only conventional NK cells but also ILC1 and NKp46⁺ ILC3, with YFP after tamoxifen administration (*J Exp Med* 2016;213:2745.). However, the population of YFP-positive conventional NK cells is higher than that of YFP-positive ILC1 and NKp46⁺ ILC3 after tamoxifen administration. Considering that the vast majority of NKp46⁺NK1.1⁺TCRβ⁻ lymphocytes in the spleen are conventional NK cells (*Nat Immunol* 2015;16:306. and *eLife* 2014;3:e01659.), ILC1 and NKp46⁺ ILC3 would be negligible in this study.

We provided a sufficient explanation of the fate-tracking system in the sections of Results and Materials and Methods (lines 98-106 and 749-753 in the main text).

3. Line 115-116: “Among three genes, we chose *Themis2* as the focus of the present study because we confirmed expression in naïve NK cells at the protein level”.

In Fig. 1b the authors say that they chose candidate genes according to them being upregulated in activated or memory NK cells relative to naïve NK cells. Is *Themis2* expressed in all NK cell subsets and is only quantitatively upregulated upon NK activation? Should the authors not validate its expression at the protein level on the relevant cells – activated and memory NK cells?

We quantified total amounts of *Themis2* protein in naïve, effector, and long-lived Ly49H⁺ and Ly49H⁻ NK cells *ex vivo* by intracellular staining of cytosolic and nuclear *Themis2* protein by flow cytometry. Naïve Ly49H⁺ and Ly49H⁻ NK cell subsets expressed equivalent amounts of *Themis2* proteins. In line with the kinetics of *Themis2* mRNA during MCMV infection (Supplementary Figures 2b and 2c), the amount of *Themis2* protein was increased in effector Ly49H⁺ NK cells and maintained in memory NK cells during MCMV infection (new Supplementary Figure 2d).

We described these results in the sections of Results, Materials and Methods, and Figure Legends in the revised manuscript (lines 126-128, 831-834, 840-843 in the main text, and lines 55-64 in Supplementary Figures).

And if *Themis2* is indeed expressed to some extent in naïve NK cells, I think it is worth discussing its possible roles in them as part of the discussion.

We agreed with the reviewer's comment. We discussed this issue in the Discussion section in the revised manuscript (lines 414-418 and 422-427 in the main text).

4. From Fig. 1 it seems like Themis2 negatively regulates NK proliferation in all stages of activation and infection. Can the authors differentiate between its initial effects and its effects in later, memory-related states?

Activation signaling via Ly49H, which triggers ZAP70/Syk phosphorylation, is essential for proliferation of Ly49H⁺ NK cells in the early course of MCMV infection (*Nature* 2009;457:557.). Our results demonstrate that Themis2 inhibits activation signaling via Ly49H by attenuating ZAP70/Syk phosphorylation (Supplementary Figure 3b), implying that Themis2 negatively regulates proliferation of Ly49H⁺ NK cells until day 7 post-infection — the peak of NK cell response after MCMV infection (Figure 1d and new Figure 1i). To clarify the effects of Themis2 on proliferation (cell division) in the contraction and memory phases, we evaluated Ki-67 in WT and *Themis2*^{-/-} Ly49H⁺ NK cells on days 7, 17, and 28 post-infection. However, Themis2 deficiency had small impact on the frequency of Ki-67⁺Ly49H⁺ NK cells during MCMV infection (new Supplementary Figure 4d). By contrast, a significantly low percentage of *Themis2*^{-/-} Ly49H⁺ NK cells expressing active Caspase 3/7 were present as compared with WT Ly49H⁺ NK cells during MCMV infection (new Figure 1g).

We described these results in the sections of Results, Discussion, Materials and Methods, and Figure Legends in the revised manuscript (lines 181-189, 423-427, 644-647, 844-861 in the main text, and lines 98-101 in Supplementary Figures).

Is it possible that Themis2 KO allows for a larger pool of NK cells at the early stages of infection, and as a result more memory NK cells can emerge?

A recent study has reported that effector Ly6C-negative Ly49H⁺ NK cells are memory precursors during MCMV infection. The authors claim that Ly6C-negative Ly49H⁺ memory precursor NK cells highly express Bcl-2 at the early stage of MCMV infection (on day 7 post-infection) and preferentially differentiate into memory NK cells (*Nat Immunol* 2022;23:556.). When we

examined the frequencies of Ly6C⁺Ly49H⁺ NK cells on day 7 post-infection, there were similar percentages of Ly6C-negative NK cells in WT and *Themis2*^{-/-} Ly49H⁺ NK cells (Supplementary Figures 4a and 5a: Percentages of Ly6C⁺Ly49H⁺ NK cells are shown). Furthermore, there were no difference in the amount of Bcl-2 in effector WT and *Themis2*^{-/-} Ly49H⁺ NK cells on day 7 post-infection (new Supplementary Figure 4e). Together, these results indicate that *Themis2* may not be actively involved in the regulation of the pool of memory precursor NK cells.

We discussed the issue in the Discussion section (lines 446-451 in the main text) and described these results in the sections of Results, Materials, Methods, and Figure Legends (lines 181-189, 850-861 in the main text, and lines 98-101 in Supplementary Figures) in the revised manuscript.

5. If *Themis2* negatively regulates NK proliferation, do *Themis2* KO mice have more naïve NK cells at baseline?

As shown in new Supplementary Figures 2f and 2g, naïve WT and *Themis2*^{-/-} NK cells did not show any significant differences in the percentage of Ly49H⁺ (receptor for MCMV m157), KLRG1⁺ (activation and differentiation marker), Ly6C⁺ (differentiation marker), DNAM-1⁺ (co-stimulatory molecule for memory NK cell differentiation) NK cells, and developmental stages of NK cells defined as CD11b and CD27 in the spleen. Furthermore, *Themis2* deficiency had no impact on the percentage of cell cycling Ki-67⁺ Ly49H⁺ NK cells in the naïve state (new Supplementary Figure 4d). These results demonstrate that *Themis2* deficiency does not affect the baseline of naïve NK cells.

6. Extended data Fig. 3 - The authors present very convincing data showing that *Themis2* negatively regulates Ly49H signaling via some mechanism involving either Zap70 or Syk phosphorylation. As this seems like a major way by which *Themis2* can regulate NK cell function, it will be beneficial to describe it a bit further and recognize the specific target affected by *Themis2*. Can this target be DAP12 and not directly Zap70 or Syk?

Among other methods, this could be performed by either repeating the experiments shown in the abovementioned figure using specific antibodies for the different proteins or by co-

immunoprecipitation experiments looking for direct interaction between Themis2 (or its Flag-tagged construct presented in fig.5), Zap70, Syk or the upstream DAP12 protein.

We appreciate the reviewer's suggestion. As the reviewer proposed, we established a human NK cell line NKL-Ly49H (DAP12⁺ZAP70⁺SYK⁻) expressing FLAG-tagged Themis2 for the immunoprecipitation (IP) of Themis2-bound proteins, because of no immunoprecipitation-validated antibody against Themis2. Cytosolic proteins from these cells were prepared after Ly49H stimulation. Because we anticipated the low binding affinity of Themis2 to DAP12/ZAP70 or the fragility of the Themis2-DAP12/ZAP70 complex, the cytosolic proteins were chemically crosslinked with DTSSP before IP of FLAG-tagged Themis2-bound proteins. After the IP, DAP12 and ZAP70 were detected in crosslinked IP products, but not non-crosslinked IP products, from Ly49H-stimulated cells (new Supplementary Figures 3d and 3e). These results suggest that Themis2 may form a complex with DAP12 and ZAP70 with a low affinity after Ly49H stimulation.

We described these results in the sections of Results, Discussion, Materials and Methods, and Figure Legends in the revised manuscript (lines 144-152, 413-414, 922-932, 1082-1094 in the main text, and lines 79-84 in Supplementary Figures).

7. Extended data fig.3 – If Themis2 negatively regulates Zap70/Syk /DAP12 signaling, can it also negatively regulate other NK receptors using a similar signaling cascade (for example NKG2D, among others)? Can the authors repeat the experiments shown in the above-mentioned figure using activating antibodies against other such receptors? In any case, a control using an antibody that activates a receptor with a different DAP12-ZAP70-Syk-independent signaling cascade is needed to further validate these results.

We appreciate the reviewer's thoughtful comment. According to the suggestion, we examined the effector functions of WT and *Themis2*^{-/-} NK cells after stimulation of NKG2D, NKp46, and NK1.1 (NKR-P1C). *Themis2*^{-/-} NK cells showed enhanced degranulation in response to these stimuli (new Supplementary Figure 3c). Thus, Themis2 inhibits activation signaling through a diverse group of activating NK receptors.

We described these results in the sections of Results, Discussion, Materials and Methods, and Figure Legends in the revised manuscript (lines 141-143, 415-419, 880-883 in the main text, and lines 77-79 in Supplementary Figures).

8. Figure 5a – Only data from activated NK cells is shown, while Themis2 appears to be translated in memory NK cells and to some extent also in naïve NK cells (Extended data fig. 2a). The differential binding and precipitation of Themis2 to proteins in these two cell stages relative to activated NK cells and to each other is also important to the understanding of Themis2's role in NK cells and memory creation.

As the reviewer pointed, the comprehensive understanding of Themis2-bound nuclear proteins in naïve, effector, and memory NK cells is important, because differential Themis2-bound proteins may control NK cell memory. However, there is a technical limitation: The quantity of nuclear protein obtained from an NK cell. We needed 2 mg nuclear proteins per sample for the pull-down and the subsequent mass spectrometry. We could obtain only 0.08 mg nuclear proteins in mouse NK cells per spleen (that is, 25 mice are needed for 2 mg nuclear proteins). Further, the frequency of MCMV-primed memory NK cells is approximately 10% in splenic NK cells in *Themis2^{-/-}* NK-CreERT2 mice (new Figures 1j). To overcome the issue, we used effector Ly49H⁺ NK cells, because they are increased in number on days 7-10 post-infection. However, it seems technically impossible to obtain an adequate amount of nuclear proteins from memory NK cells. Although we were interested in Themis2-bound nuclear proteins in naïve, effector, and memory NK cells comprehensively, we used only effector NK cells for these procedures.

We could mention the technical limitation in the Results or Discussion section, if necessary.

Technical issues

1. Lines 69-71: unclear sentence: “They recognize unhealthy cells by utilizing a repertoire of activating NK receptors and exert the effector function, i.e., cytotoxicity, interferon (IFN)- γ production, for the eradication of these cells”.

Should it be their effector functions instead? Also, it can be understood from this sentence that NK cells only secrete IFN γ , which is of course only a single example of the several cytokines and chemokines they secrete as part of their effector functions.

We apologize for the sentence. We corrected the sentence (lines 66-67 in the main text).

2. Line 79: "Ly49H⁺ NK cells expand as effector NK cells, undergo apoptosis after the peak of NK cell response, and differentiate into long-lived memory NK cells during MCMV infection".

The sentence might be clearer if it will be "...undergo apoptosis after the peak of NK cell response, or differentiate into long-lived memory NK cells"

According to the reviewer's suggestion, we corrected the sentence (line 75 in the main text).

3. Extended data figure 2a,4a – The authors show a representation of 1 experiment. Can they also show the quantification of all repeats?

We quantified these results and provided the graphs with these representative flow cytometric plots of Supplementary Figures 2a and 4a. We also add them in the sections of Material and Methods and Figure Legends in the revised manuscript (lines 830, 840-843 in the main text and lines 50-51, 94-95 in Supplementary Figures).

4. Fig. 3c+d: The cells described in figure 3a+b are human. Are the cells depicted in Fig. 3c+d human or mouse? How were they generated? These results would be clearer if this information will be presented in the text and/or in the figure legend.

We apologize for the inadequate explanation. The cells used in Figures 3a and 3b is a human NK cell line transfectant expressing mouse Ly49H. On the other hand, NK cells in Fig 3c and 3d are primary mouse NK cells. Naïve Ly49H⁺ NK cells were purified from the spleen of tamoxifen-administered *Themis2^{+/+}* and *Themis2^{-/-}* NK-CreERT2 mice on day 10 after tamoxifen administration. Effector and memory Ly49H⁺ NK cells were purified from the spleen of MCMV-infected and tamoxifen-administered *Themis2^{+/+}* NK-CreERT2 mice on days 10 and 25 after tamoxifen administration.

We added the explanation in the sections of Results and Figure Legends (lines 232-235, 671, and 676-680 in the main text).

Responses to the Reviewer #3

In their manuscript, Shibuya and colleagues provide evidence that Themis2 acts as a negative regulator of NK cell responses against MCMV infection. Using mass spec, they mechanistically determine a novel themis2 binding partner Zfp740, generate luciferase reporter lines to show Zfp740 is a repressor, and then make new Zfp740 ko mice to recapitulate the Themis2 phenotype. These impressive findings will be of broad interest to those in the field of immunology. I feel that with the edits I've suggested below, this manuscript would be a great fit for publication in Nature Communications.

Specific comments:

1. Is NK cell effector function (e.g., IFN γ production/degranulation) similarly enhanced in response to stimulation by other activating receptors (e.g., anti-NK1.1, anti-NKG2D, and/or anti-Ly49D)? Given that the hypothesis that Themis2 interaction may only be specific for activating receptors that utilize DAP12 as co-receptors, these data would add nuance the overall findings.

We appreciate the reviewer's comment. We examined effector functions of WT and *Themis2*^{-/-} NK cells after stimulation of NKG2D, NKp46, and NK1.1 (NKR-P1C). *Themis2*^{-/-} NK cells showed enhanced degranulation in response to these stimuli (new Supplementary Figure 3c). Thus, Themis2 inhibits activation signaling through a diverse group of activating NK receptors.

We described these results in the sections of Results, Discussion, Materials and Methods, and Figure Legends in the revised manuscript (lines 141-143, 415-419, 880-883 in the main text, and lines 77-79 in Supplementary Figures).

2. Themis2 is highly upregulated in effector and memory Ly49H⁺ NK cells compared to naïve cells. Is the enhanced effector function in these states associated with enhanced pZAP70/Syk as well? Do the authors see differences in IFN γ production on day 2 PI in vivo?

According to the reviewer's comment, we examined IFN- γ production and activation of WT and *Themis2*^{-/-} Ly49H⁺ NK cells on day 1.5 post-infection. Consistent with enhanced phosphorylation of ZAP70/SYK in *Themis2*^{-/-} Ly49H⁺ NK cells, *Themis2*^{-/-} Ly49H⁺ NK cells displayed a more activated phenotype and produced larger amounts of IFN- γ than WT Ly49H⁺ NK cells on day 1.5 post-infection (new Supplementary Figures 3f and 3g).

We described these results in the sections of Results, Materials and Methods, and Figure Legends in the revised manuscript (lines 153-159, 769-770 in the main text, and 84-89 in Supplementary Figures).

A discussion on how the Calcium-NFAT signaling would be affected and contribute to the overall enhanced response in *Themis2*-deficient NK cells would be of interest.

We appreciate the reviewer's comment. According to the reviewer's suggestion, we added the discussion on how the Calcium-NFAT signaling would be affected by *Themis2* (lines 403-411 in the main text).

3. While the authors implied enhanced survival in *Themis2*-deficient Ly49H⁺ NK cells, they have not demonstrated that the expansion and contraction advantage observed in these cells are due to cell survival versus proliferation. Assessing Ki-67, Bcl2, Bim, and/or Caspase3/7 at different time points during infection could provide further proofs that this phenotype is due to enhanced survival. CTV can also be used to assess differences in proliferation.

We appreciate the reviewer's comment. To clarify the effects of *Themis2* on proliferation (cell division), survival, and apoptosis in the contraction and memory, we evaluated Ki-67, Bcl-2, Bim, and active Caspase 3/7 in WT and *Themis2*^{-/-} Ly49H⁺ NK cells on days 7, 17, and 28 post-infection. After the peak of NK cell response, a significantly low percentage of *Themis2*^{-/-} Ly49H⁺ NK cells expressing active Caspase 3/7 were present as compared with WT Ly49H⁺ NK cells after MCMV infection (new Figure 1g). By contrast, *Themis2* deficiency had small impact on the frequency of Ki-67⁺ NK cells and expression of Bim and Bcl-2 in Ly49H⁺ NK cells during MCMV infection (new Supplementary Figures 4d-f). These results indicate that *Themis2* negatively regulates Caspase 3/7-mediated apoptosis in the contraction and memory phases.

We described these results in the sections of Results, Discussion, Materials and Methods, and Figure Legends in the revised manuscript (lines 181-189, 423-427, 644-647, 844-861 in the main text, and lines 98-101 in Supplementary Figures).

Minor:

1. How many genes showed differential expression between WT and *Themis2*-deficient Ly49H⁺ NK cells in each time point and pairwise comparison? Can PCA analysis be shown and included in the supplementary figure?

We performed the PCA and also compared the number of differentially expressed genes (DEGs) in individual modules of naïve Ly49H⁺ *Themis2*^{+/+} and *Themis2*^{-/-} NK cells and memory *Themis2*^{+/+} and *Themis2*^{-/-} NK cells.

We provided these results as new Supplementary Figures 6a and 6b (PCA and pairwise heatmap, respectively) and added them in the sections of Results, Materials and Methods, and Figure Legends in the revised manuscript (lines 251, 984, 1025-1026 in the main text and 122-125 in Supplementary Figures).

2. How many peaks were identified from H3K4me3 and H3K27Ac ChIP-seq between WT and *Themis2*-deficient NK cells at naïve and memory time points?

After the peak calling, the following number of ChIP-seq peaks were identified.

- Naïve *Themis2*^{+/+} NK cells: 18,183-19,745 H3K27ac; and 14,109-18,345 H3K4me3
- Naïve *Themis2*^{-/-} NK cells: 16,466-20,510 H3K27ac; and 14,656-15,501 H3K4me3
- Memory *Themis2*^{+/+} NK cells: 14,756-20,764 H3K27ac; and 14,529-17,703 H3K4me3
- Memory *Themis2*^{-/-} NK cells: 18,204-19,323 H3K27ac; and 13,301-16,113 H3K4me3

We provided the information in the Materials and Methods section in the revised manuscript (lines 996-999 in the main text).

3. What do the authors mean by “pairwise intersection analyses”? Does it refer to comparing overlapping histone peaks between WT and *Themis2*-deficient NK cells? Was a unified atlas of all conditions used to generate this Enrichment heatmap plot? Or does it utilize individual atlas?

If the latter, then please use the unified atlas instead of individual atlas. This might help reveal the differences in H3K4me3 and H3K27Ac signals.

We apologize for the misleading sentence. “Pairwise intersection analyses” means visualization of ChIP-seq peaks by pairwise intersections (pairwise intersection heatmaps) as an easy-to-interpret visual approach. We corrected the term in the sections of Results and Materials and methods in the revised manuscript (lines 263 and 1014 in the main text).

We assessed overlaps/ differences in chromatin landscapes among naïve and memory *Themis2*^{+/+} and *Themis2*^{-/-} NK cells by pairwise intersections of H3K4me3 and H3K27ac ChIP-seq peaks. We used individual atlases of H3K4me3 and H3K27ac ChIP-seq data (Figure 4d). According to the reviewer’s suggestion, we used the unified atlas of H3K4me3 and H3K27ac ChIP-seq data. The pairwise intersection heatmap revealed that *Themis2* has little impact on the overall correlations in H3K4me3 and H3K27ac regions in all NK cell subsets (new Supplementary Figure 7h).

We described these results in the sections of Results, Materials and Methods, and Figure Legends in the revised manuscript (lines 286-289, 1014-1015 in the main text, and 146-149 in Supplementary Figures) .

4. Are the differentially expressed genes (DEGs) between WT and *Themis2*-deficient memory NK cells the ones that exhibit differences in H3K4me3 signals between WT and *Themis2*-deficient NK cells? Can the authors provide some examples and global integrative analysis of H3K4me3 and RNA-seq?

According to the reviewer’s suggestion, we combined RNA-seq and H3K4me3 ChIP-seq data of memory *Themis2*^{+/+} and *Themis2*^{-/-} NK cells and performed an integrative analysis for the correlation between gene expression of memory DEGs and signals of memory H3K4me3 differentially accessible regions (DARs). The scatter plots suggested the possibility that *Themis2* might be involved in gene expression and promoter accessibility of DNA-template transcription-associated genes (new Supplementary Figure 7i).

We described these results in the sections of Results, Materials and Methods, and Figure Legends in the revised manuscript (lines 290-293, 1029-1035 in the main text, and 149-

152 in Supplementary Figures).

5. What is the rationale for overlapping CTCF binding sites with H3K27Ac? How was the CTCF binding atlas generated? Is it based on scanning the mouse genome with CTCF motifs? Or is it from NK cell CTCF ChIP-seq? If so, from what time point?

We apologize for the inadequate explanation. When we performed GO analysis of memory H3K27ac DAR genes, we could not find any GO terms with statistical significance. We speculated the reason why H3K27ac regions are not only found in enhancers/ enhancer-like elements, but also in promoters. To extract “non-promoter H3K27ac-marked regions”, we overlapped our H3K27ac ChIP-seq data with public CTCF-bound enhancer signature (ELS) datasets available in ENCODE (reference 59, *Nature* 2020;583:699.). CTCF was first described as a transcriptional repressor (*Oncogene* 1990;5:1743), but was also found to act as a transcriptional activator (*Mol Cell Biol* 1993;13:7612.). Through the literature, CTCF ChIP-seq analyses revealed that CTCF binding correlates with enhancer activity and enhancer-promoter interaction to maintain gene expression (reference 60, *Mol Cell* 2017;67:1049.). Thus, overlapping H3K27ac ChIP-seq data and CTCF-bound ELS can highlight non-promoter H3K27ac-marked regions.

We added the rationale in the sections of Results and Materials and Methods (lines 279-280 and 1004-1007 in the main text).

6. How many proteins are pulled down by this assay? And how many of those are not transcription factor? Given that Themis1/2 contain Proline-rich sequence (PRS) that binds to SH3 domain-containing proteins, such as Grb2, do the authors observed any of SH3 domain-containing proteins getting pulled down?

We detected 35 proteins by the pull-down and the subsequent mass spectrometry. Among them, 15 transcription factors and 20 non-transcription factors were identified. We further chose 9 transcription factors that were sufficiently expressed in NK cells (Supplementary Figure 8d). We could not find any SH3 domain-containing proteins among these Themis2-bound nuclear

proteins, which was confirmed by referring to UniProt and SMART databases, probably because we used nuclear proteins (not cytosolic proteins) of mouse NK cells.

We added the information on how many proteins were detected in the Supplementary Figure 8d.

REVIEWERS' COMMENTS

Reviewer #1 (Remarks to the Author):

I am satisfied that the authors have addressed the points I raised in my initial review, as well as adding substantial mechanistic detail in response to queries from the other reviewer. With respect to my own queries, the new experiments do highlight that the role of Themis is not specific to memory NK cells: it would be helpful if a sentence were added to the abstract to convey this.

Reviewer #2 (Remarks to the Author):

1. There might be a typo in one of the added sentences: Lines 447-451, the problematic section is bolded & underlined:

"In the present study, the populations of WT and Themis2^{-/-} Ly6C⁻ Ly49H⁺ NK cells showed comparable each other on day 7 pi (Supplementary Fig. 4a, 5a), and WT and Themis2^{-/-} Ly49H⁺ NK cells equivalently expressed Bcl-2 during MCMV infection (Supplementary Fig. 4e), suggesting that Themis2 may not be involved in the regulation of the size of the memory precursor NK cell pool."

2. Regarding the authors question in their reply to comment #8:

Mentioning this in the discussion section can be useful for future studies, as this is indeed an interesting, relevant and open question that emerges from this study, and such a comment might lead to an answer in future studies using other methods. Even so, it is not a critical issue and the manuscript is of high quality and relevance with or without it, so it is completely up to the authors.

Reviewer #3 (Remarks to the Author):

The authors have adequately addressed all of my comments in the revised manuscript and I believe this insightful study is now ready for publication.

Responses to the Reviewer #1

I am satisfied that the authors have addressed the points I raised in my initial review, as well as adding substantial mechanistic detail in response to queries from the other reviewer. With respect to my own queries, the new experiments do highlight that the role of Themis is not specific to memory NK cells: it would be helpful if a sentence were added to the abstract to convey this.

We appreciate the reviewer's comment. In the present study, we principally focused on the role of Themis2 in NK cell memory function and formation. Instead of adding the sentence in the Abstract, we mentioned the role of Themis2 in the effector function of NK cells in the Discussion section (lines 417-420).

Responses to the Reviewer #2

1. There might be a typo in one of the added sentences: Lines 447-451, the problematic section is bolded & underlined:

"In the present study, the populations of WT and Themis2^{-/-} Ly6C⁻ Ly49H⁺ NK cells showed comparable each other on day 7 pi (Supplementary Fig. 4a, 5a), and WT and Themis2^{-/-} Ly49H⁺ NK cells equivalently expressed Bcl-2 during MCMV infection (Supplementary Fig. 4e), suggesting that Themis2 may not be involved in the regulation of the size of the memory precursor NK cell pool."

We thank the reviewer for this comment. We are afraid we can't find the typo pointed out by the reviewer due to the disappearance of the bold and underline, but Ly6C⁻ should be Ly6C⁺.

2. Regarding the authors question in their reply to comment #8:

Mentioning this in the discussion section can be useful for future studies, as this is indeed an interesting, relevant and open question that emerges from this study, and such a comment might lead to an answer in future studies using other methods. Even so, it is not a critical issue and the manuscript is of high quality and relevance with or without it, so it is completely up to the authors.

We appreciate the reviewer's comment. We mentioned the topic and the technical limitation in the Discussion section (lines 434-445).